# How does a warm and low–snow winter impact the snow cover dynamics in a humid and discontinuous boreal forest? Insights from observations and modeling in eastern Canada

Benjamin Bouchard[1,2,3], Daniel F. Nadeau[1,2], Florent Domine[3,4,5], François Anctil[1,2], Tobias Jonas[6], Étienne Tremblay[1]

[1]Department of Civil and Water Engineering, Université Laval, Quebec City, G1V 0A6, Canada
[2]CentrEau – Water Research Centre, Université Laval, Quebec City, G1V 0A6, Canada
[3]Centre d'Études Nordiques, Université Laval, Quebec City, G1V 0A6, Canada
[4]Department of Chemistry, Université Laval, Quebec City, G1V 0A6, Canada
[5]Takuvik Joint International Laboratory, Université Laval (Canada) and CNRS-INSU (France), Quebec City, G1V 0A6, Canada
[6] WSL Institute for Snow and Avalanche Research (SLF), 7260 Davos Dorf, Switzerland

*Correspondence to*: Benjamin Bouchard (benjamin.bouchard.1@ulaval.ca)

**Abstract.** In the boreal forest of eastern Canada, winter temperatures are projected to increase substantially by 2100. This region is also expected to receive less solid precipitation, resulting in a reduction in snow cover thickness and duration. These changes are likely to affect hydrological processes such as snowmelt, the soil thermal regime, and snow metamorphism. The exact impact of future changes is difficult to pinpoint in the boreal forest, due to its complex structure, and the fact that snow dynamics under the canopy are very different from those in the gaps. In this study, we assess the influence of a low–snow and warm winter on snowmelt dynamics, soil freezing, snowpack properties, and spring streamflow in a humid and discontinuous boreal catchment of eastern Canada (47.29° N, 71.17° W, ≈ 850 m AMSL) based on observations and SNOWPACK simulations. We monitored the soil and snow thermal regimes and sampled physical properties of the snowpack under the canopy and in two forest gaps during an exceptionally low–snow and warm winter, projected to occur more frequently in the future, and during a winter with conditions close to normal. We observe that snowmelt was earlier but slower, top soil layers were cooler, and gradient metamorphism was enhanced during the low–snow and warm winter. However, we observe that snowmelt duration increased in forest gaps, that soil freezing was enhanced only under the canopy, and that snow permeability increased more strongly under the canopy than in either gap. Our results highlight that snow accumulation and melt dynamics are controlled by meteorological conditions, soil freezing by forest structure, and snow properties by both weather forcing and canopy discontinuity. Overall, observations and simulations suggest that the exceptionally low spring streamflow in W20–21 was mainly driven by low snow accumulation, slow snowmelt and low precipitation in April and May rather than enhanced percolation through the snowpack and soil freezing.

# 1 Introduction

The boreal forest is one of the most extensive biomes on Earth. It is projected to warm by up to 5°C by 2100, with the largest increases occurring in winter (Scheffer et al., 2012; Zheng et al., 2023; Price et al., 2013; IPCC, 2022). Warmer winters will result in less solid precipitation and in a thinner, shorter–lived snow cover (Laternser and Schneebeli, 2003; Hamlet et al., 2005). Spring melt will occur earlier in the season, but at a slower rate because less radiative energy is available at that time (López-Moreno et al., 2013; Musselman et al., 2017). Together, these changes are expected to reduce peak spring streamflow and runoff volumes (Furey et al., 2012; Berghuijs et al., 2014; Luce and Holden, 2009; Barnhart et al., 2016). Projections for the boreal forest of eastern Canada, characterized by humid and cold conditions in winter (D'Orangeville et al., 2016; Isabelle et al., 2020), point towards an increase in winter streamflow and an earlier spring freshet with more snow accumulation in the north and less in the south (Guay et al., 2015). The interannual variability of precipitation and temperature is also projected to increase, making warm and dry winters more likely (Ouranos and MELCCFP, 2022).

It has been shown that the ground thermal regime is strongly influenced by the amount of snow accumulation (Zhang, 2005; Slater et al., 2017). In forests, the spatial pattern of soil temperatures is difficult to determine because snow depth is highly variable (Mellander et al., 2005). Observations from a subalpine forest plot in Switzerland show that frost penetrates the ground deeper under tree crowns due to less snow accumulation than in forest gaps which reduces the infiltration and increases surface runoff (Stadler et al., 1996). Infiltration is also limited during low–snow winters due to a thinner snowpack that favors soil freezing (Hardy et al., 2001; Shanley and Chalmers, 1999). It is clear that both canopy structure and snow conditions influence the ground thermal regime, soil freezing and infiltration. However, it is not well understood which of these two factors predominates over the other because they have not been investigated simultaneously in a single study.

Forest structure affects not only the dynamics of soil freezing, but also the physical properties of the snowpack. As canopy interception limits snow accumulation under trees (Pomeroy et al., 1998; Mazzotti et al., 2019; Sun et al., 2018), stronger vertical temperature gradients ($\partial T/\partial z$) favor kinetic snow grain growth under the canopy more than in gaps (Albert and Hardy, 1993; Molotch et al., 2016; Bouchard et al., 2022). Bouchard et al. (2022) showed that the phenomenon results in lower specific surface area (SSA) and greater snowpack permeability under the canopy. With warmer winters, one may expect snow surface temperature to increase, but snow thickness to decrease. Domine et al. (2007) have suggested a decrease in permeability with climate warming due to a warmer snow surface, lower $\partial T/\partial z$, and slower grain growth. The authors also noted that an increase in the frequency of melting events would result in more melt–freeze crusts and low–permeability ice layers (Albert and Perron, 2000). In the boreal forest, where snow surface temperature and snow accumulation are highly variable in space (Malle et al., 2019; Parajuli et al., 2020; Mazzotti et al., 2023b) and where the canopy structure influences crust formation in the snowpack (Bründl et al., 1999; Teich et al., 2019; Bouchard et al., 2022), the impacts of warmer winters on snow properties may be hard to predict. Therefore, dedicated studies on the influence of warmer winters on snowpack physical properties in discontinuous boreal forests are needed.

The main research gap that motivates our work is that winter weather conditions and canopy structure have not been studied together to see how they influence snowmelt dynamics, the ground thermal regime, and the physical properties of the snowpack. Thus, the objective of this study is to quantify the effect of a low–snow and warm winter on the aforementioned processes in a humid and discontinuous boreal forest. To assess this, we compared snow melt, snow physical properties, soil freezing, and spring runoff at a small catchment in the south part of the humid boreal forest of eastern Canada, for two consecutive winters. One winter was exceptionally warm and dry, while the other was slightly colder, with precipitation amounts similar to the standard climatology of the study region. These contrasted conditions represent an ideal comparison to investigate some expected effects of climate change. Extensive snow monitoring and pit measurements, supported by multi-layer snowpack simulations under the canopy, were conducted to achieve the research objective.

Section 2 presents the study site, the instrumentation, the observed and estimated physical variables, and the modeling setup. In Sect. 3, we present the climatology of each winter and the differences in melt rate, snowpack and ground thermal regime, and the evolution of snow cover physical properties within medium–size and small forest gaps and under the canopy. The simulated water content profile and runoff from the snowpack and the measured spring streamflow of the catchment are also presented for both winters in Sect. 3. Finally, we compare our results with existing literature, present the limitations of the study, and discuss the potential climatic, hydrological, and ecological implications of warmer winters in a discontinuous humid boreal forest in Sect. 4.

## 2 Methods

### 2.1 Study site

The study took place in the Montmorency Forest (MF) during winters 2020–2021 (W20–21) and 2021–2022 (W21–22). Measurements began on 15 October and ended on 15 June of the following year. MF is located in the province of Québec, in eastern Canada, at the southern edge of the boreal forest (47.29° N, 71.17° W; Fig. 1a).

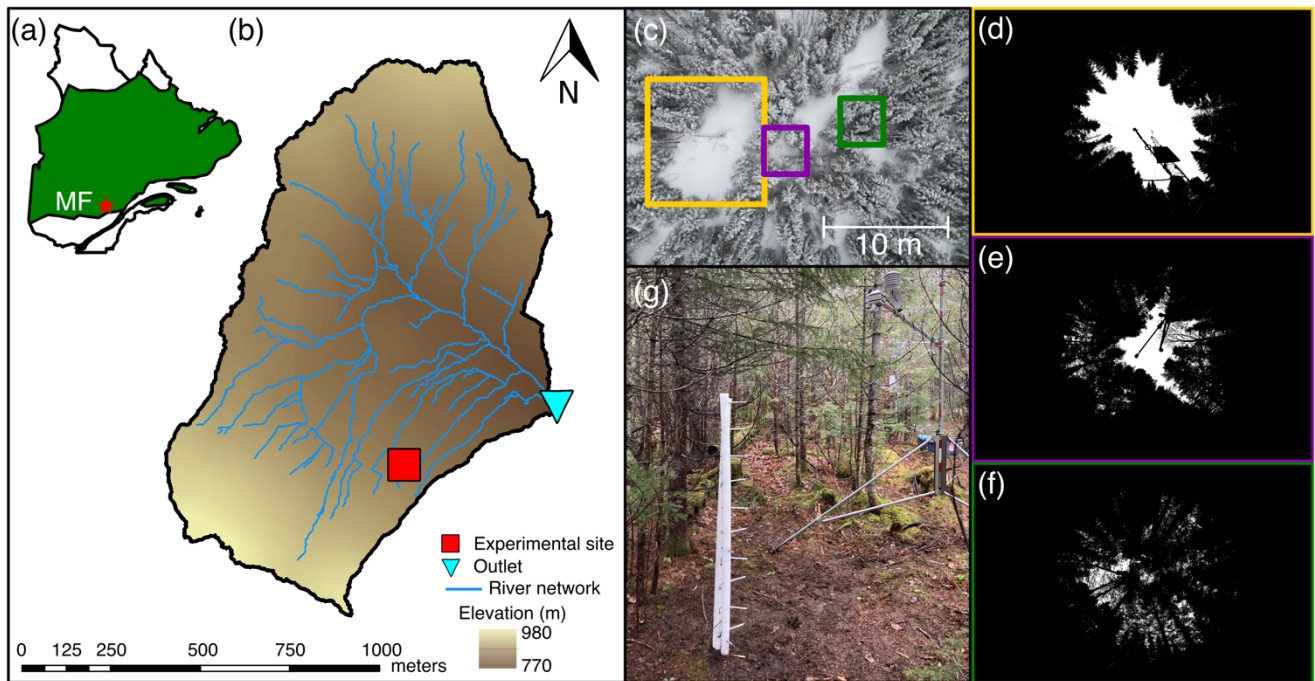

Figure 1: Map of the province of Quebec, Canada, with the location of the Montmorency Forest (MF) indicated by a red star and the boreal zone in green (a). Elevation map of the study catchment (BEREV-7A) with the location of the experimental site and the outlet of the catchment (b). Aerial view of the experimental site with the medium gap (yellow), the small gap (purple), and the canopy (green) stations (c) and black and white hemispherical photos of each station location (d–e–f). Picture of the monitoring station under the canopy (g).

Three monitoring stations were installed in a 1.1 km$^2$ forested subcatchment (7A) of the *Bassin expérimental du Ruisseau des Eaux–Volées* (BEREV) located at MF (Fig. 1b). The stations were installed in a medium gap, a small gap, and under the canopy (Fig. 1c). Table 1 presents the gap dimensions and the sky–view fraction (SVF) at each site. The SVF was evaluated using an adaptive thresholding algorithm (Jonas et al., 2020) on hemispherical photographs taken with a Sigma 8mm f3.5 EX DG Circular Fisheye lens in the fall of 2022 (Fig. 1d–e–f). Stations were located on a 12°, northeast–facing slope at 846 m above mean sea level (AMSL), within a 10 m tall stand of balsam fir (*Abies balsamea*) mixed with white birch (*Betula papyrifera*) and white spruce (*Picea glauca*) trees. The catchment is a regeneration from a major logging operation that took place in 1993 (Guillemette et al., 2005). The soil is a sandy loam topped by ≈7 cm of litter. The stations were located within 30 m of a 20 m flux tower measuring shortwave (0.3–2.8 μm) and longwave (4.5–42 μm), upwelling and downwelling radiation (CNR4; Kipp & Zonen). Given the small size of the catchment and the location of the stations close to the average elevation of the catchment, we assume that the snow measured at the experimental site is representative of the entire catchment. A V–notch streamflow gauge and a bubble flowmeter, maintained by the provincial government and in operation since 1967, are located at the outlet of the catchment (DEH station 051004, https://www.cehq.gouv.qc.ca/hydrometrie/historique_donnees/). Based on LiDAR imagery, we estimate that 75 % of the catchment is canopy–covered, with the remainder being gaps of small and medium size. Four kilometers northeast of BEREV-7A, there is a 0.01 km$^2$ open area at 664 m AMSL, the NEIGE site (Pierre et al., 2019),

which hosts a federal weather station (ECCC station 7042395). We used data from the federal station to position the winters of 20–21 and 21–22 in relation to the MF climatology. The NEIGE site also hosts a CS725 instrument (Campbell Scientific) that monitors the snow water equivalent (SWE) at a 6 h timestep, using differential gamma–ray absorption (Choquette et al., 2013) and an ultrasonic snow height sensor (Judd Communication) working on an hourly timestep.

| Stations | Gap dimensions (m × m) | Sky−view fraction (0 –1) |
|:---:|:---:|:---:|
| **Medium gap** | 10 × 14 | 0.26 |
| **Small gap** | 2 × 3 | 0.10 |
| **Canopy** | - | 0.08 |

Table 1: Gap dimensions and sky–view fraction of each station

## 2.2 Measurements of physical variables

In this study, the monitoring stations tracked the thermal regime of the snowpack and of the top 20 cm of soil, the soil volumetric water content (VWC) and the effective thermal conductivity of the snow ($k_s$) at two levels in the snowpack. Measurements from the stations were complemented with snow pit observations of snow density ($\rho_s$), temperature, SSA, and a visual identification of the snowpack stratigraphy conducted periodically during both winters.

### 2.2.1 Monitoring Stations

Each station includes a snow height sensor, either the Judd model or the SR50a from Campbell Scientific, mounted 3 m above the ground, and is equipped with a vertical array of PT1000 thermistors (Schneider Electric). Temperature was measured at depths of 20, 10 and 5 cm in the soil, at the ground surface and then every 15 cm in the snowpack until a maximum height of 180 cm. Thermistors were inserted into white–painted aluminum tubes held in place by a vertical UHMW plastic rod. Snow surface temperature was measured at each station with an SI-111 infrared radiometer (Apogee Instruments) mounted at the same level as the snow height sensor. Each station was also equipped with an HMP60 (Campbell Scientific) sensor that measured air temperature and relative humidity at a height of 3 m above the ground. A CS655 reflectometer (Campbell Scientific) measured the VWC at a depth of 15 cm into the soil. Variations in 15 cm VWC were used as a proxy for infiltration. We also define the onset of snowmelt as the beginning of snowpack runoff, when the soil VWC reaches a maximum in spring. A CR10X data logger (Campbell Scientific) recorded point measurements every hour. Also, time–lapse cameras taking hourly photos were used to visually identify the precipitation phase and the burying of the thermistors. An example of a monitoring station, the one under the canopy in this case, is presented in Fig. 1g.

The stations were also equipped with two heated TP08 needle probes (NP; Hukseflux) to measure $k_s$ following Morin et al. (2010) and Domine et al. (2015). The lower NP was installed 10 cm above the ground at each station, whereas the higher NP was installed 65 cm above the ground under the canopy, and 80 cm in the forest gaps to account for the greater snow height in these environments. To prevent snow melting around the heated needle during measurements, these were taken only every other day between 5:00 and 6:00 AM, only when the snow temperature was below –2.5°C. During each measurement, the NP

was heated for 150 s with a power of 0.45 W m$^{-1}$. The heating curve was recorded on a CR1000 data logger (Campbell Scientific). The algorithm developed to automatically assess $k_s$ from the heating curves is derived from Domine et al. (2015) and is described in the Supplementary Material (Fig. S1). The error associated with snow thermal conductivity measurements with these static needle probes is estimated to be 21 % (Domine et al., 2015).

We performed detailed soil profile measurements at the canopy and small gap sites on 13 and 20 July 2021, respectively. At the canopy site only, we measured soil thermal conductivity at different depths using a Hukseflux TP02 heating needle probe. Soil temperature was measured every 5 cm from the surface to 30 cm below and every 10 cm down to 80 cm below the surface with a Greinsinger Pt-1000 temperature probe (resolution: 0.1°C). Soil cores ($\approx$ 165 cm$^3$) were taken from each layer, which were then weighted before and after oven drying for 48 hours at 65 °C and 100 °C for organic and mineral soils, respectively. This allowed to estimate the volumetric water content and the bulk density of the soil, assuming a density of water of 1000 kg m$^{-3}$. Figure S2 presents the vertical profiles of soil characterization at both locations.

### 2.2.2 Data gap filling

Small data gaps (1 to 5 h) in the snow height and temperature time series were filled by linear interpolation. We used data from the other stations to fill the longer data gaps, as described in the Supplementary Material (Fig. S3 – S5). This approach was applied to snow height in the medium forest gap (37 days in 2020) and under the canopy (77 days in 2022), to snow surface temperature in the small forest gap (winter 2021–22), and to air and snow surface temperatures under the canopy (16 days in 2022).

### 2.2.3 Snow pit measurements

Four snow pits were dug in medium gaps, small gaps and under the canopy each winter, for a total of 24 snow pits over the two winters. Snow pits in gaps and under the canopy were all dug at sites with similar conditions (gap size, SVF) to the monitoring stations, all within 150 m of the stations. Table 2 lists the date of each snow pit measurement.

For each snow pit survey, we measured $\rho_s$ with a Snow-Hydro 100 cm$^3$ box with a ±10% accuracy (Conger and McClung, 2009). Snow density was measured every 3 to 5 cm in the vertical direction. In the presence of ice columns, measurements were taken adjacently. We measured snow temperature using the Greinsinger Pt-1000 probe every 5 cm in the topmost 40 cm of snow and every 10 cm for the lower snow layers. The SSA, being a quantitative indicator of snow metamorphism (Taillandier et al., 2007), was measured using the DUFISSS instrument (Gallet et al., 2009). DUFISSS uses infrared reflectance of snow samples with an integrating sphere at 1310 nm to estimate the SSA with an accuracy of approximately 12% (Gallet et al., 2009). Measurements were taken vertically every 1 to 5 cm depending on the stratigraphy.

| | Medium gap | Small gap | Canopy |
|---|---|---|---|
| **Winter 2020–21** | 8 Dec. 20 | 8 Dec. 20 | 8 Dec. 20 |
| | 27 Jan. 21 | 27 Jan. 21 | 26 Jan. 21 |
| **(low−snow)** | 9 Mar. 21 | 10 Mar. 21 | 10 Mar. 21 |
| | 6 Apr. 21 | 6 Apr. 21 | 6 Apr. 21 |
| **Winter 2021–22** | 13 Jan. 22 | 12 Jan. 22 | 12 Jan. 22 |
| | 15 Feb. 22 | 15 Feb. 22 | 15 Feb. 22 |
| **(reference)** | 10 Mar. 22 | 11 Mar. 22 | 10 Mar. 22 |
| | 28 Apr. 22 | 29 Apr. 22 | 28 Apr. 22 |

**Table 2: Snow pit measurement dates**

## 2.3 Estimated variables

### 2.3.1 Snow water equivalent and snowpack cold content

We estimated the SWE (m) from snow density profiles using:

$$SWE = \frac{h\overline{\rho_s}}{\rho_w},$$
(1)

where $h$ and $\overline{\rho_s}$ are respectively the height (m) and the average density (kg m$^{-3}$) of the snowpack, and $\rho_w$ is the density of liquid water (1000 kg m$^{-3}$).

The snowpack cold content ($Q_{CC}$, in J m$^{-2}$) is the quantity of energy needed to bring the snowpack to its melting point. It is defined as:

$$Q_{CC} = -c_{ice} \sum_{i=1}^{n} h_i \overline{\rho_{s,i}} \left(\overline{T_{s,i}} - T_m\right),$$
(2)

where $c_{ice}$ is the heat capacity of ice (2108 J kg$^{-1}$ K$^{-1}$), $h_i$ and $\overline{\rho_{s,i}}$ are the same as above but for a given snow layer $i$. $\overline{T_{s,i}}$ is the average temperature between the bottom and top boundaries of each $i$ layer (K) as monitored at the stations and $T_m$ is the melting point of ice (273.15 K). $h_i$ corresponds to the vertical distance between each temperature measurement.

Since snow density profiles were point measurements, we interpolated $\overline{\rho_{s,i}}$ linearly between each snow pit date at an hourly timestep to match temperature monitoring. Snow density was assumed constant from the beginning of the snow season until the date of the first snow pit (23 days in W20–21 and 59 days in W21–22) and then from the last snow pit date until the melt–out (39 days in W20–21 and 31 days in W21–22). To validate this rough hypothesis, we measured four density profiles in W20–21 and five in W21–22 at NEIGE site next to the SWE sensor and compared the resulting estimated SWE time series with direct observations, as shown in the Supplementary Material (Fig. S6).

### 2.3.2 Snowpack net shortwave and longwave radiation

We used the HPEval model (Jonas et al., 2020) to estimate the downwelling shortwave radiation below the canopy ($SWR_{\downarrow,bc}$)
from the hemispherical photographs taken at the monitoring stations and the incoming shortwave radiation measured above
the canopy at the flux tower, some 10 m away. HPEval combines hemispherical imagery of the canopy and radiation transfer
modeling to estimate subcanopy shortwave radiation at very high spatial and temporal resolution. Reflected shortwave
radiation by the snowpack ($SWR_{\uparrow,bc}$) was estimated using five arbitrary albedo classes defined from the work from Hardy et
al. (2000) and Melloh et al. (2001). The five classes are listed in Table 3.
We manually assigned a daily albedo class to the snow surface by visually inspecting timelapse photographs from each station.
A sample representative photo of each albedo class is presented in Fig. S7. The net shortwave radiation below canopy
($SWR_{net,bc}$) is the difference between $SWR_{\downarrow,bc}$ and $SWR_{\uparrow,bc}$.

| Class | Albedo | Description |
|-------|--------|-------------|
| 1 | 0.80 | Dry snow |
| 2 | 0.70 | Dry snow with litter and/or rough surface |
| 3 | 0.65 | Wet snow |
| 4 | 0.55 | Wet snow + litter |
| 5 | 0.40 | Wet snow + lots of litter |

Table 3: Albedo classes used for the estimation the reflected shortwave radiation by the snow surface

We estimated the downwelling longwave radiation below the canopy ($LWR_{\downarrow,bc}$; W m$^{-2}$) by using:

$$195 \quad LWR_{\downarrow,bc} = SVF \times LWR_{\downarrow,ac} + (1 - SVF)\,\varepsilon_{can}\,\sigma\,T_{can}{}^4 , \tag{3}$$

where $LWR_{\downarrow,ac}$ is the downwelling longwave radiation measured above the canopy at the nearby flux tower (W m$^{-2}$), $\sigma$ is the
Stefan–Boltzmann constant ($5.67 \times 10^{-8}$ W m$^{-2}$ K$^{-4}$), $\varepsilon_{can}$ is the canopy emissivity, set to 0.98 (Pomeroy et al., 2009), and $T_{can}$
is the canopy temperature (K). We assume that the canopy temperature can be approximated by the air temperature measured
at our monitoring stations.
The upwelling longwave radiation below the canopy ($LWR_{\uparrow,bc}$; W m$^{-2}$) is determined with:

$$LWR_{\uparrow,bc} = \varepsilon_s\,\sigma\,T_{surf}{}^4 , \tag{4}$$

where $\varepsilon_s$ is the emissivity of the snow surface, set to 0.99 (Sicart et al., 2006), and $T_{surf}$ is the snow surface temperature (K)
measured at each station. The net longwave radiation below canopy ($LWR_{net,bc}$) is the difference between $LWR_{\downarrow,bc}$ and
$LWR_{\uparrow,bc}$. The net total radiation below canopy ($R_{net,bc}$) is the sum of $SWR_{net,bc}$ and $LWR_{net,bc}$.

### 2.3.3 Ground heat flux

The ground heat flux was assumed equivalent to the snow heat flux within the lower 15 cm of the snowpack (Lackner et al., 2022), and calculated using Fourier's law (Eq. 5):

$$F = -k_s \frac{(T_{15} - T_0)}{dh},$$ (5)

where $F$ (W m$^{-2}$) is the heat flux of the lower 15 cm of the snowpack, $k_s$ (W m$^{-1}$ K$^{-1}$) is the effective thermal conductivity of this basal snow layer, taken as the NP measurement at a 10 cm height, $T_{15}$ and $T_0$ are the temperatures measured 15 cm above the ground and at the ground surface (K), respectively, $dh$ is the thickness of the bottom layer (15 cm). To avoid melting of the ice matrix, NPs were not heated when the snow was warmer than –2.5°C. Because of this constraint, $k_s$ at 10 cm from each station was only measured at the beginning of the winter on both years when the bottom of the snowpack was colder than –2.5°C. This resulted in more than 80% $k_s$ measurements that were missing. In order to have complete time series of $k_s$, we used Eq. 18 from Fourteau et al. (2021) with our pit density measurements to estimate $k_s$. We validated the use of this empirical equation by comparing it to our observations of pit density and monitoring of $k_s$ at 10 and 65 or 80 cm at all stations. We used $k_s$ that was measured the closest in time to the snow pit surveys. Before comparing our observations with the equation, correction factors between 1.1 and 1.3, depending on snow temperature, were applied to $k_s$ measurements to account for the systematic underestimation of the fixed NP approach (Fourteau et al., 2022). Doing this, we found a correlation coefficient of 0.70 and a bias of 10.2% between our observations and the equation from Fourteau et al. (2021). Comparison between the corrected measurements and the equation is detailed in the Supplementary Material (Fig. S8).

### 2.3.4 Vertical temperature gradient

The magnitude of the mean snowpack vertical temperature gradient ($|\partial T / \partial z|$; in °C m$^{-1}$) is expressed as follow:

$$|\partial T / \partial z| = \left| \frac{(T_{surf} - T_0)}{h_s} \right|,$$ (6)

where ($T_{surf} - T_0$) is the temperature difference between the snow surface and the soil–snow interface (in °C) and the $h_s$ is the snow height (in m).

### 2.3.5 Snow permeability

Snow permeability ($K_s$; in m$^2$) indicates the ease with which a fluid subjected to a pressure gradient flows through a porous medium. $K_s$ can be estimated from $\rho_s$ and the optical grain radius ($r_g$; in m) according to the following equation (Calonne et al., 2012):

$$K_s = 3 r_g{}^2 e^{-0.013 \rho_s}.$$ (7)

We assume that each snow layer is a collection of independent ice spheres ($\rho_{ice}$ = 917 kg m$^{-3}$) with the same SSA as the snow of interest (Grenfell and Warren, 1999). Therefore, $r_g$ is estimated as follow:

$$r_g = \frac{3}{\rho_{ice}SSA}. \tag{8}$$

## 2.4 Modeling setup

We simulate the snowpack using the SNOWPACK model, version 3.6.0 (Lehning et al., 2002), coupled with the two-layer canopy module implemented by Gouttevin et al. (2015) for winter 20-21 and winter 21-22. SNOWPACK is a multilayer snow model that solves Richards equations for liquid water transport in the snowpack (Wever et al., 2014). The two-layer canopy scheme has shown a reasonably good performance in simulating the thermal inertia of the canopy and the underlying snowpack (Gouttevin et al., 2015; Todt et al., 2018; Bouchard et al., 2024). The SNOWPACK canopy module does not simulate the heterogeneous structure of the canopy, so snow-forest processes within forest gaps are not parameterized in the model. Therefore, we used the simulations described in Bouchard et al. (2024), referred to by the authors as the "Initial Module", for winters 20-21 and 21-22 at the MF site.

Briefly, simulations were performed at a 15 min time step using local meteorological forcing data measured above the canopy thanks to a 20 m flux tower and recorded at a 30 min time step (Isabelle et al., 2018). We also used measurements from a double fence automatic reference for precipitation inputs (Pierre et al., 2019). Based on field measurements, the tree height and the leaf area index (LAI) were set to 9.2 m and 4.8 m$^2$ m$^{-2}$, respectively. The stand basal area was set to 0.005 m$^2$ m$^{-2}$ based on Hadiwijaya et al. (2020). The direct throughfall fraction was set to 0 to better represent the subcanopy snowpack. We used values from Gouttevin et al. (2015) for the canopy albedo and the two-layer LAI fraction. Finally, the initial soil parameterization was based on field measurements taken in the summer of 2021. Additional details on the forcing data, the initial canopy and soil parameterization, and SNOWPACK initialization file are given in the Methods section of Bouchard et al. (2024). Note that the authors found a good agreement between the simulations and observations for snow cover height and duration, snow surface temperature and snow density profiles using this modeling setup. This demonstrates that the model is suitable for simulating the canopy snowpack at the MF site.

## 3 Results

### 3.1 Climatic conditions

W20–21 was the driest winter of the 1982–2022 period, with 199 mm recorded from January to April (JFMA), including 167 mm of solid precipitation (Fig. 2; Table 4). This corresponds to a precipitation anomaly of –224 mm (–53%). In comparison, the JFMA anomaly of precipitation in W21–22 was only +5 mm (+ 1%). December–January–February (DJF) temperature in W20–21 was also 2.8°C warmer than the 1982–2022 average. As for DJF, W20–21 was the fourth warmest year of the last 40 years. In comparison, DJF temperature was 1.8°C colder in W21–22 than the 1982–2022 average and ranks as the sixth coldest

for that period. The exceptionally dry and warm winter of 2020–21 resulted in the earliest melt–out in the last 40 years at NEIGE site (11 April 2021), with snow disappearing 23 days earlier than the 1982–2022 average (Table 4). In comparison, the projected DJF temperature and the total JFMA precipitation are expected to be –10.4 °C and 425 mm, respectively, by

265 2070 at the Montmorency Forest, based on the SSP5-8.5 emission scenario from CMIP6 climate simulations (ClimateData.ca, 2023). Two heavy rain-on-snow (ROS) events were observed in December 2020, three smaller ones in November and December 2021, and several others in March, April, and May of both years. As the data show, the winter 2020–21 received a much lower than average solid precipitation and was significantly warmer resulting in a thin snowpack. We will refer to it as our low–snow and warm winter. The precipitation and temperature anomaly in the winter of 2021–22 is much weaker.

Therefore, W21–22 will correspond to the reference winter for the rest of the analysis.

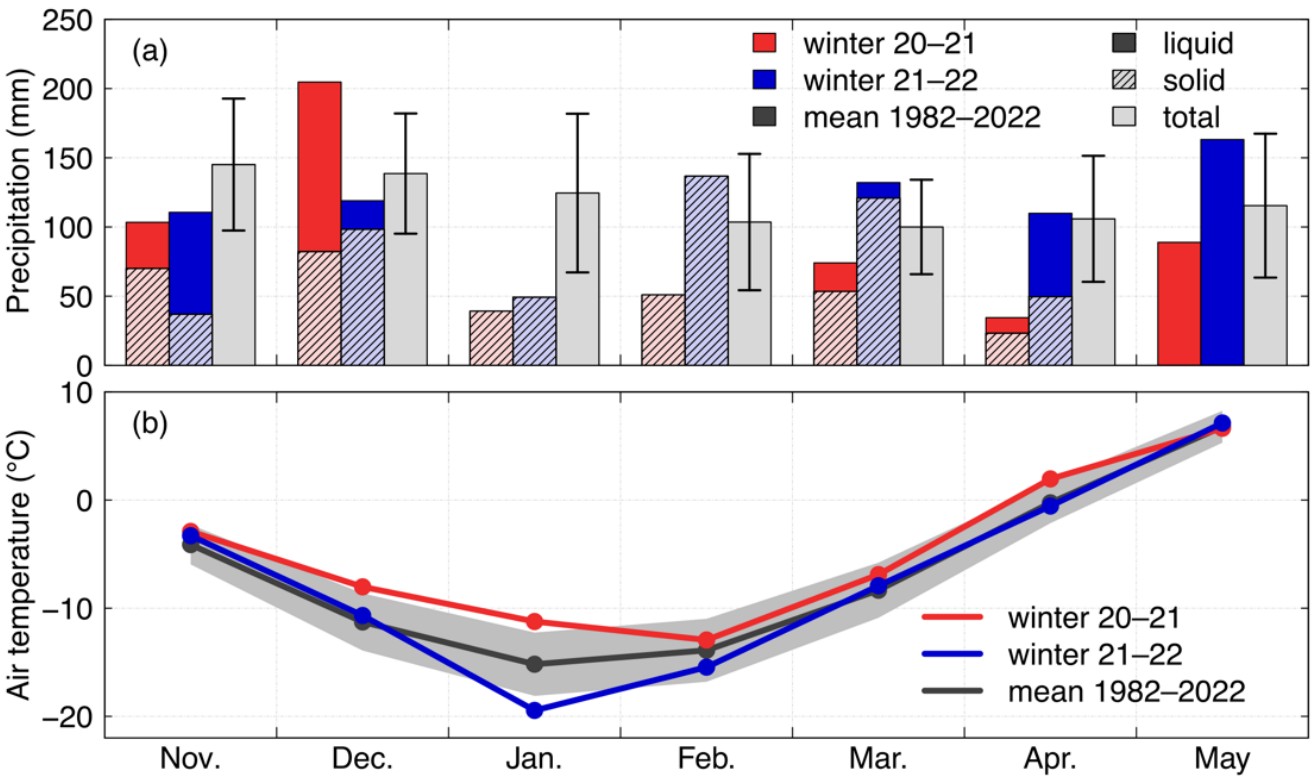

**Figure 2: Monthly sum of precipitation (a) and means of air temperature (b) from November to May in the winters of 2020–21 and 2021–22, measured at the NEIGE site and compared to the monthly sum of precipitation and means of hourly temperature over the 1982–2022 period. Dashed bars in (a) show snowfall whereas solid bars indicate rainfall. The standard deviation of the 1982–2022 is**
275 **shown by the black error bars for precipitation in (a) and by the gray area for the monthly temperature in (b).**

| | Winter 2020–21 | | | Winter 2021–22 | | |
|---|---|---|---|---|---|---|
| | value | anomaly | rank (out of 40)* | value | anomaly | rank (out of 40)* |
| $P_{tot}$ JFMA (mm) | 199 | –224 | 1 | 428 | +5 | 16 |
| $P_{tot}$ (Nov-May) | 596 | –237 | 2 | 821 | +12 | 19 |
| $P_{sol}$ (Nov-May) | 364 | –198 | 2 | 596 | +13 | 19 |
| Temp. DJF (°C) | –10.7 | +2.8 | 4 | –15.2 | –1.8 | 31 |
| $H_{s,max}$ (cm) | 67 | –38 | 5 | 142 | +37 | 33 |
| Melt–out day (DOY) | 101 | –23 | 1 | 134 | +11 | 33 |

* Temp. DJF is ranked in descending order, whereas all the other variables are ranked in ascending order.

**Table 4: Total precipitation from January to April (JFMA) and from November to May, solid precipitation from November to May, mean December–January–February (DJF) temperature, maximum recorded snow height ($H_{s,max}$), and the melt–out day at the NEIGE site for the 2020–21 and 2021–22 winters. The anomaly relative to the 1982–2022 period is also shown, along with the rank of both winters for each metric relative to the 40 winters of the climatology. We used a threshold at 1°C to define the precipitation phase over the analysis period. Winters 1999 to 2003 were excluded of the analysis because of too many missing data.**

## 3.2 Snow accumulation and melt dynamics

Due to a lower snowfall, the maximum snow height ($H_{s,max}$) in W20–21 was on average 44% lower than in the reference winter (Fig. 3). As the air was warmer in W20–21, the snow surface temperature (SST) was on average 1.35 °C warmer than in the reference winter (Fig. 4). In contrast, because there was less snow on the ground in W20–21, heat transfer through the snow was facilitated, resulting in the base of the snowpack being colder in W21–22. Due to canopy interception, less snow accumulated under the trees than in the gaps in both years and the $H_{s,max}$ was on average 35% lower under the canopy than inside both gaps. Interestingly, topmost snow layers seem to be colder under the canopy than inside gaps. This is in contradiction with Fig. 4, which shows that the *SST* was higher under the canopy than inside gaps in both years (+1.53 °C), despite similar air temperature at all three stations. This discrepancy is further discussed in Sect. 4.2.

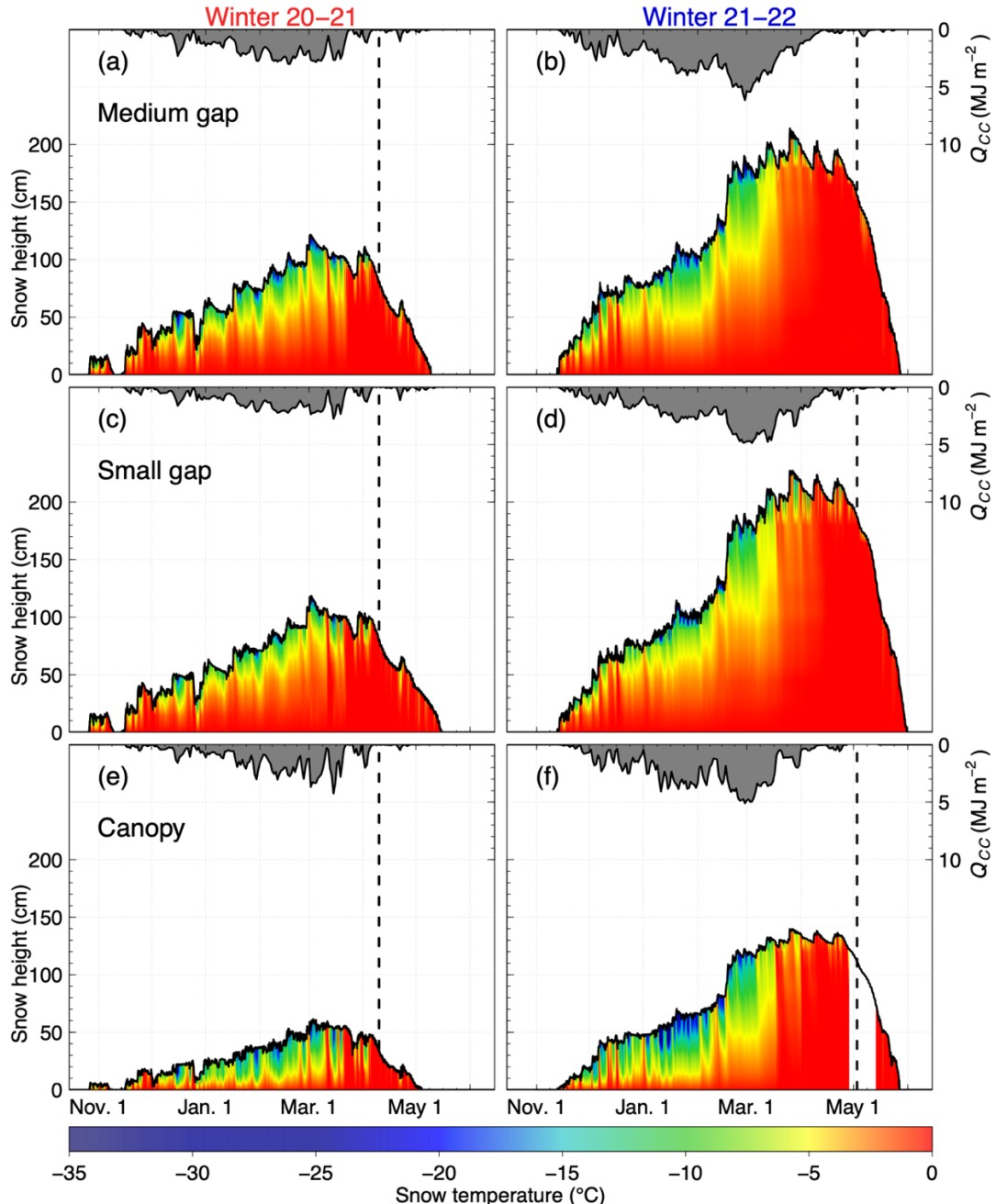

Figure 3: Snowpack thermal regime and cold content in the medium gap (a–b), in the small gap (c–d) and under the canopy (e–f) for winters 2020–21 (left) and 2021–22 (right). Dashed lines show the onset of snowmelt.

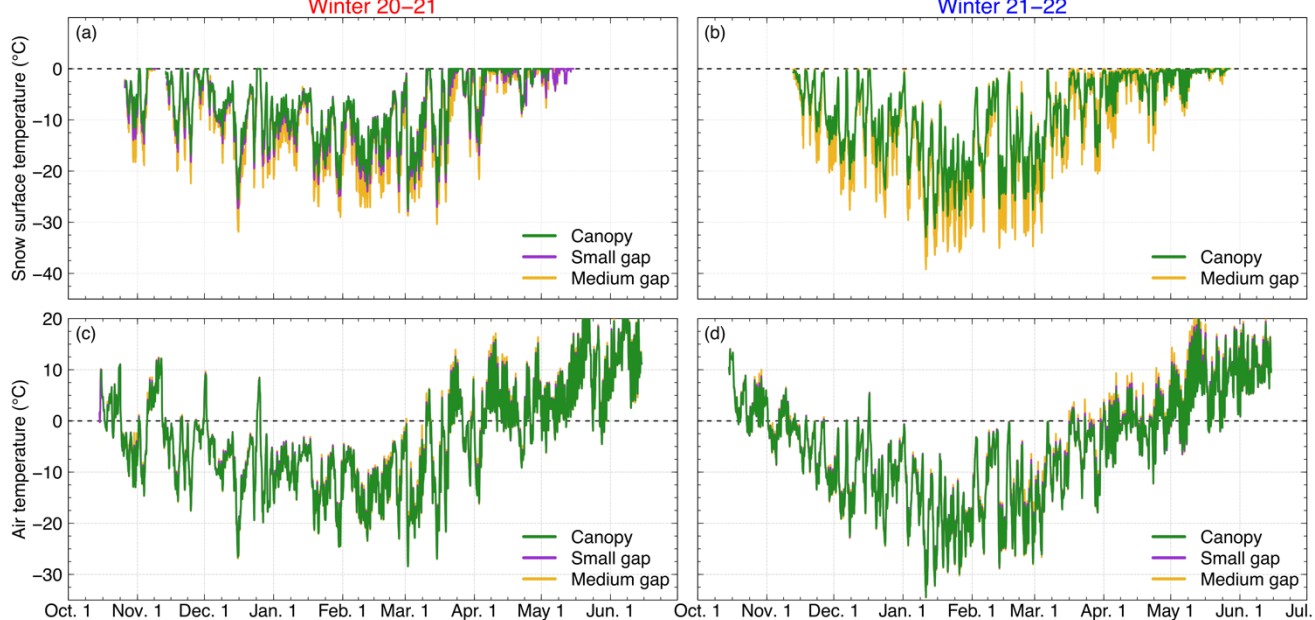

**Figure 4: Snow surface and air temperature measured in the medium gap (yellow), in the small gap (purple) and under the canopy (green) during winters 2020–21 (a–c) and 2021–22 (b–d). Since the snow surface temperature in the small gap was entirely gap–filled in winter 21–22, it is not presented in (b).**

Due to the thinner snowpack and warmer air temperature, the $Q_{CC}$ was on average 36% lower in W20–21 than in W21–22. Interestingly, the $Q_{CC}$ peaked in late February in both winters. However, since less energy was required to warm the snowpack to 0°C, snowmelt started on average 23 days earlier in the low–snow and warm winter than in the reference winter (10 April 2021 vs 3 May 2022), which is a substantial difference. Since the snowpack was thicker in the gaps than under the canopy, the $Q_{CC}$ was slightly less under the trees than in both gaps.

As snowmelt started more than three weeks earlier in W20–21 than in the reference winter, less radiative energy was available to contribute to snowmelt (Fig. 5). Net shortwave ($SWR_{net,bc}$) and longwave ($LWR_{net,bc}$) radiation below canopy were both lower in the low–snow and warm year. As expected, we observed a decrease in $SWR_{net,bc}$ and an increase in $LWR_{net,bc}$ as the sky–view fraction decreased regardless of the year.

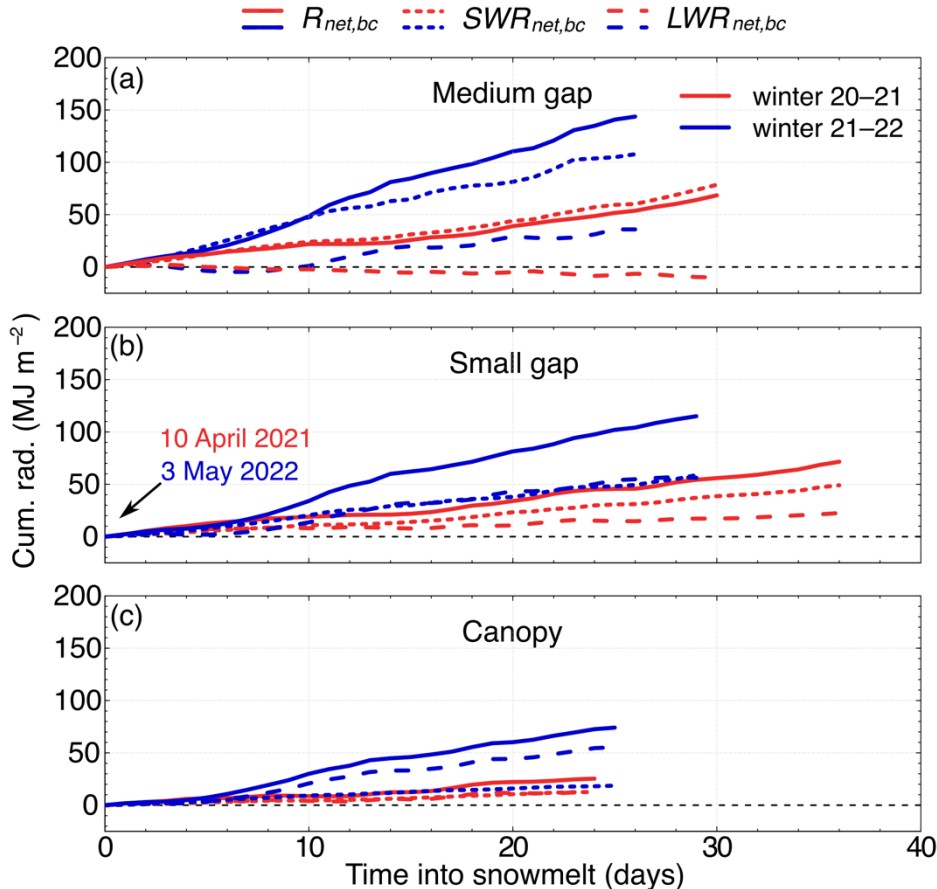

**Figure 5: Cumulative net total ($R_{net,\,bc}$), shortwave ($SWR_{net,\,bc}$) and longwave ($LWR_{net,\,bc}$) radiation below the canopy during snowmelt of winter 2020–21 (red) and 2021–22 (blue) in the medium gap (a), in the small gap (b) and at the canopy station (c). Graphs start at the beginning of the snowmelt period on 10 April 2021 in the first year and on 3 May 2022 in the second year. Note that $SWR_{net,\,bc}$ and $LWR_{net,\,bc}$ overlap during the snowmelt of 2021.**

In the medium gap, small gap and under the canopy, the SWE at the beginning of snowmelt in the low–snow year was 56%, 59% and 76% lower, respectively, than in the reference year (Fig. 6). In addition to a thinner snowpack, the melt rate was also significantly smaller during the low–snow winter. From W20–21 to W21–22, the duration of the melt period slightly increased from 26 to 31 days and from 29 to 37 days in the medium and small gaps, respectively, whereas it did not change under the canopy (24 days). The melt-out date was on average 19 days earlier in the warm year for all sites. In comparison, the difference in melt-out date between the canopy and gap stations was much smaller, with snow melting on average 5 days earlier under the canopy in both years.

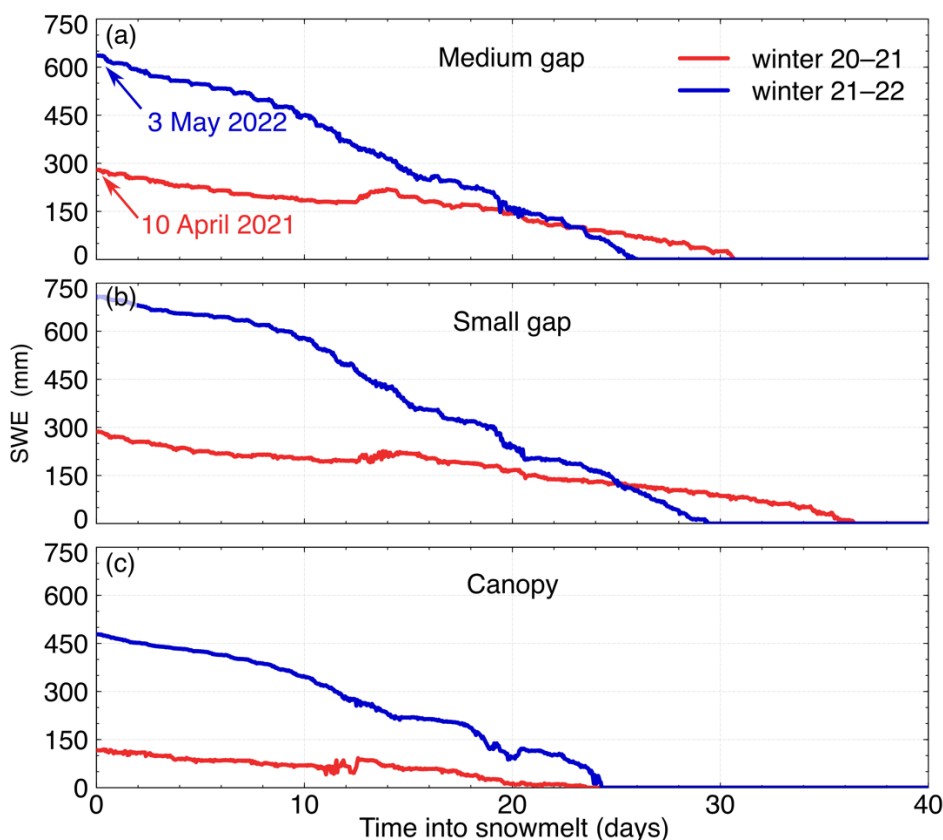

**Figure 6: SWE evolution during snowmelt of winters 2020–21 (red) and 2021–22 (blue) in the medium gap (a), in the small gap (b) and under the canopy (c). The plots start at the beginning of the snowmelt period on each year, which differs by an average of 23 days between seasons.**

### 3.3 Ground thermal regime and water content

The ground heat flux (GHF) in DJF was on average 50% higher in W20–21 than in the reference winter (Fig. 7). The largest difference was observed under the canopy, where the GHF was significantly larger than in gaps in W20–21 and in W21–22. At any given depth, the soil was cooler in W20–21 than in W21–22 in DJF (Fig. 7; Table 5), even though it was a warmer season. In both winters, the top few centimeters of soil below the canopy dropped below freezing, while all soil layers in the gaps remained above or at 0°C. We also observed a frozen soil–snow interface during snow pit surveys under the canopy, but never in the gaps. In the low–snow and warm winter, the freezing depth reached 10 cm, compared to only 5 cm in the reference winter, even though it was much colder. However, the soil–snow interface was below 0°C for similar durations (103 days vs 102 days).

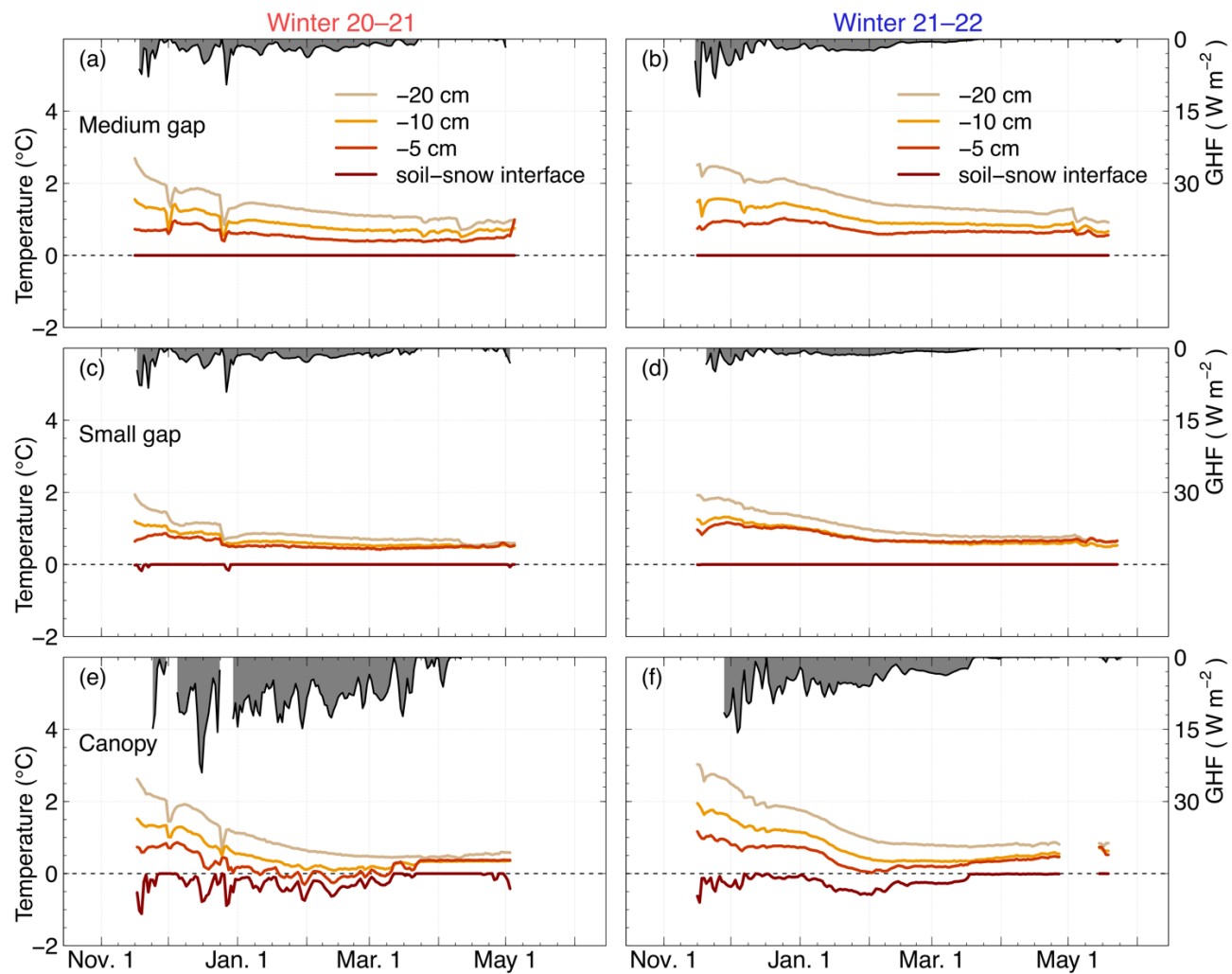

**Figure 7:** Ground heat flux (GHF) and temperature at 20, 10, 5 cm below ground level and at the soil–snow interface during the snow cover period of winters 2020–21 (left) and 2021–22 (right) in the medium gap (a–b), in the small gap (c–d) and under the canopy (e–f).

| | Stations | | | | | |
|---|---|---|---|---|---|---|
| | **Medium gap** | | **Small gap** | | **Canopy** | |
| **Soil Depth** | Temp. 20–21 (°C) | ΔT (°C) | Temp. 20–21 (°C) | ΔT (°C) | Temp.20–21 (°C) | ΔT (°C) |
| **Soil–snow interface** | 0 | 0 | 0 | 0 | −0.32 | −0.05 |
| **5 cm** | 0.59 | −0.20 | 0.55 | −0.29 | 0.15 | −0.28 |
| **10 cm** | 0.92 | −0.23 | 0.67 | −0.21 | 0.49 | −0.32 |
| **20 cm** | 1.40 | −0.35 | 0.88 | −0.31 | 1.00 | −0.42 |

**Table 5:** Mean December–January–February temperature at the soil–snow interface and at 5, 10 and 20 below ground level and at the soil–snow interface in the medium gap, in the small gap and under the canopy for winter 20–21 and the difference with winter 21–22.

Figure 8 shows that soil volumetric liquid water content (VWC) at 15 cm depth was higher under the canopy than in gaps for both years, except in spring and after two heavy ROS that occurred early in W20–21. Soil profile characterization that was performed in a small gap and under the canopy during the summer of 2020 showed different porosity in the topmost 30 cm of soil (Fig. S2) which could explain the differences in VWC over a short distance at our study site. On 10 April 2021 (Fig. 8a), the increase in VWC was sharp and sudden in both gaps, while it was more gradual under the canopy. A data acquisition error occurred at the canopy station in April and May 2022 (Fig. 8b), so we cannot compare the increase in 15 cm VWC under the canopy at the onset of snowmelt in W20–21 and W21–22. In the medium and small gaps, we observed a similar behavior of the 15 cm VWC at the beginning of snowmelt in 2022.

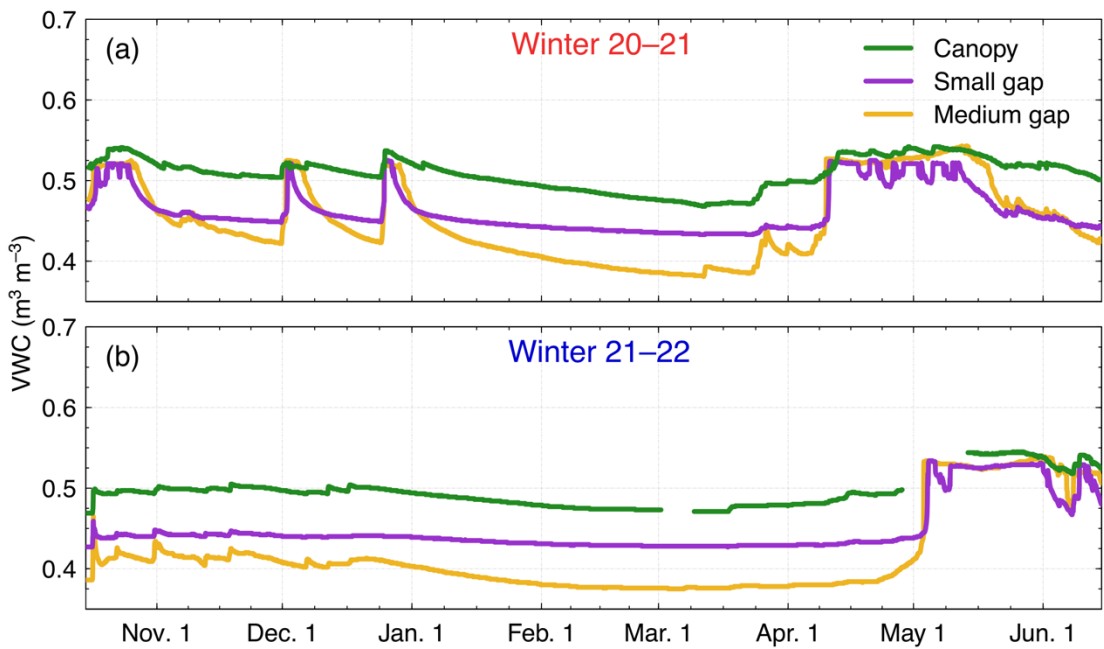

**Figure 8: Soil liquid water content (VWC) at 15 cm below the surface in the medium gap (yellow), in the small gap (purple) and under the canopy (green) in winters 2020–21 (a) and 2021–22 (b).**

### 3.4 Vertical temperature gradient and snow properties

From November to January, the average $|\partial T/\partial z|$ was similar at each site and year, except under the canopy where the gradient was much larger during the warm year (Fig. 9). In February and March of the low–snow and warm winter, $|\partial T/\partial z|$ remained within or above the transition zone from equilibrium to kinetic crystal growth (Colbeck, 1983). In contrast, in February of the reference winter, the decrease of $|\partial T/\partial z|$ was more intense and we observed a drop of $|\partial T/\partial z|$ below 10°C m$^{-1}$ in gaps. $|\partial T/\partial z|$ was also higher under the canopy than in gaps, in particular before February where the canopy-gaps difference is much stronger.

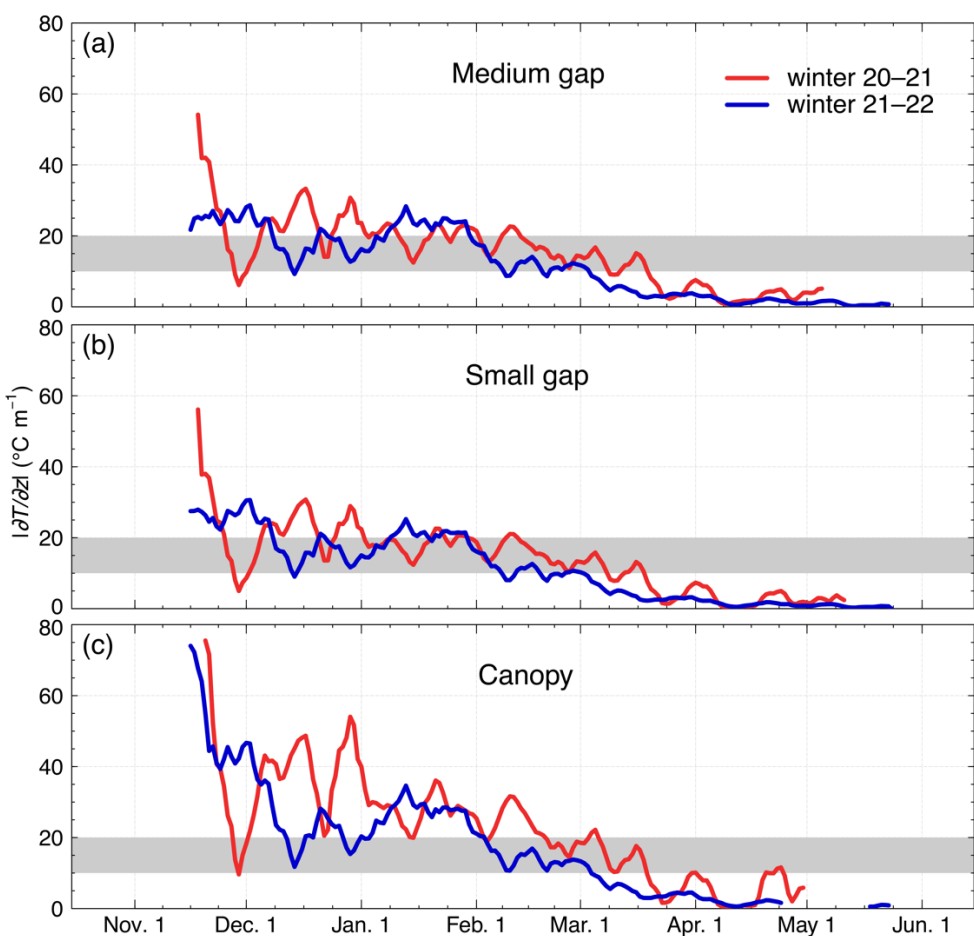

**Figure 9: 7 day rolling mean of the magnitude of the vertical temperature gradient ($|\partial T/\partial z|$) in winters 2020–21 (red) and 2021–22 (blue) in the medium gap (a), the small gap (b) and under the canopy (c). The grey band on each frame shows the transition zone from equilibrium to kinetic growth.**

Figure 10 presents the comparison of snow stratigraphy, density, SSA and permeability between both years for gaps and under

the canopy as obtained from snow pit observations between 9 and 11 March both years. These snow pit dates are convenient

to present as they are similar in both years and correspond to a snowpack under dry snow conditions and well into the snow

accumulation period. Observations from the other snow pits are presented in Supplementary Material (Fig S9 – S12).

At all sites, the top of the faceted crystals layer (FC) rises higher in the snowpack in the warm and low–snow year than during

the reference year. In winter 20–21, this height includes FC and depth hoar (DH), as well as a thick layer of melt–freeze

polycrystals (MFpc) resulting from the December 2020 ROS. In this basal layer, we also observed FC and DH but these are

secondary to clusters of polycrystals. As a result, we observed fewer rounded grains (RG) in W20–21 than in W21–22. In both

years, the combination of FC and DH layers was proportionally thicker under the canopy than in the gaps and the thickness of

the DH was noticeably higher under the canopy. Overall, the level of faceting is the greatest in the snowpack under the canopy

during the warm and low–snow year.

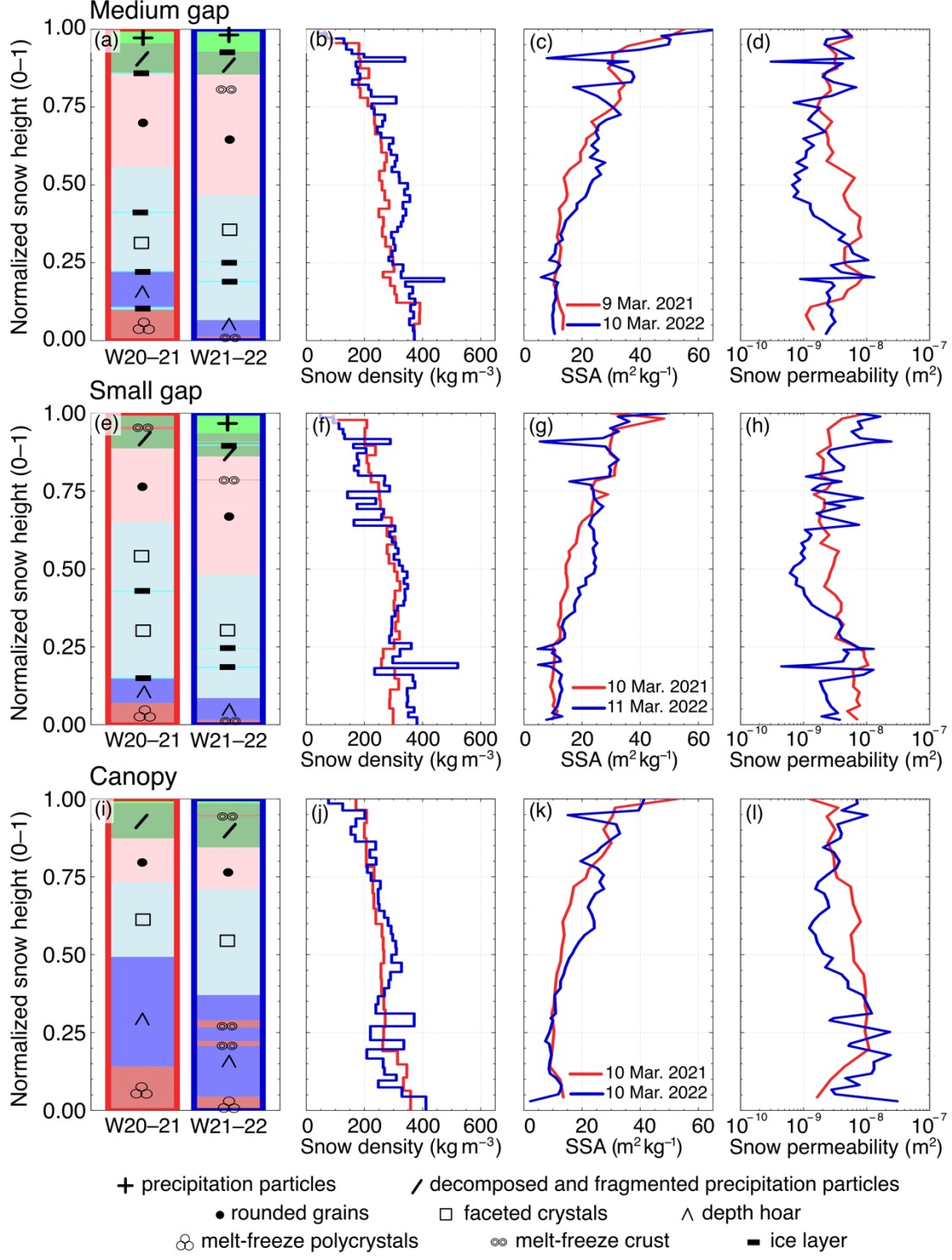

**Figure 10: Stratigraphy, snow density, SSA, and snow permeability profiles as measured in the medium gap (a–b–c–d), in the small gap (e–f–g–h) and under the canopy (i–j–k–l) on 9 and 10 March 2021 (red) and 10 and 11 March 2022 (blue).**

Snow density was lower in W20–21 than in the reference winter (257 kg m$^{-3}$ versus 276 kg m$^{-3}$ on average). In general, the density profiles showed values increasing with depth and were lower under the canopy than in gaps in both winters. Kruskal–Wallis tests performed on density measurements showed that the density difference between both winters at all sites was significant, as well as the canopy-gaps density difference in both winter (p-value < 0.05).

The vertical profile of SSA followed the expected general shape, with higher values near the surface (fresh snow) and lower values deeper (FC and DH). In the bottom 25% of the snowpack, there was no significant difference in SSA between both winters (p-value < 0.05). This may be due to the ice and melt–freeze layers SSA measurement technique that was improved from W20–21 to W21–22. Removing the ice and melt–freeze layers from the analysis leads to a significantly greater SSA in W21–22, coherent with a lower temperature gradient. At a normalized height of 0.25 to 0.75, where the transition from RG to FC occurred, the SSA was 19% lower during the low–snow winter than in the reference winter, which was statistically significant (p-value < 0.05). As a result of a lower density and SSA, the snow permeability was 57% higher in the warm and low–snow winter within the height range 0.25 to 0.75. This difference between both years was also statistically significant, as demonstrated by a Kruskal–Wallis test (p-value < 0.05). Note that in both years, the SSA and the snow permeability was lower and higher, respectively, under the canopy than in gaps for the middle half of the snowpack. This difference was also significant based on a Kruskal–Wallis test (p-value < 0.05).

### 3.5 Spring streamflow

In 2021, air temperatures became positive in early April, allowing a decline in SWE. In 2022, this occurred 20 days later (Fig. 11a–b). The daily melt rate was much smaller in the first winter than in the second, as already shown in Fig. 6. In April and May, rain-on-snow accounted for 54 mm in 2021 compared to 202 mm the following year (Fig. 11c–d).

Consistent with our observations, the simulated subcanopy snowpack was much thinner at the onset of snowmelt in 2021 than in 2022 (Fig. 11e–f). The wetting front simulated by SNOWPACK in winter 20–21 took 57 h at an average rate of 30 cm d$^{-1}$ to reach the ground in early April, while it took 149 h at 26 cm d$^{-1}$ in early May in the following year. The model also simulated a thick basal ice layer in W20–21 and a thin ice layer deep in the snowpack in W21–22, which is also in line with snow pit observations (Fig. 10i).

Besides modeling a much earlier snowpack runoff, the model simulated spring runoff with quite different patterns in both years (Fig. 11g–h). In 2021, SNOWPACK generated intermittent runoff driven by several episodes of refreezing of the entire snow column, while in 2022 the simulations resulted in a continuous runoff throughout the snowmelt period, with peaks driven by liquid precipitation. The runoff simulations in both years are generally consistent with the observations of discharge at the outlet of BEREV-7A catchment. In 2021, the measured discharge was generally lower than the simulated runoff and slightly delayed. This was also the case in April 2022. However, during the snowmelt of 2022, the discharge measurements were similar to or greater than the simulated snowpack runoff in 2022, and the observed and simulated runoff peaks were synchronized.

Overall, low snowmelt and low liquid precipitation in the spring of the first year resulted in significantly lower spring streamflow runoff than in the second years. In April and May 2021, the average runoff was 3.1 mm d$^{-1}$ compared to 8.5 mm d$^{-1}$ in the following year. The spring runoff volume from the low–snow and warm winter was the lowest volume observed at the outlet of BEREV-7A for April and May since discharge monitoring began in 1968. In 2022, it was the sixth highest.

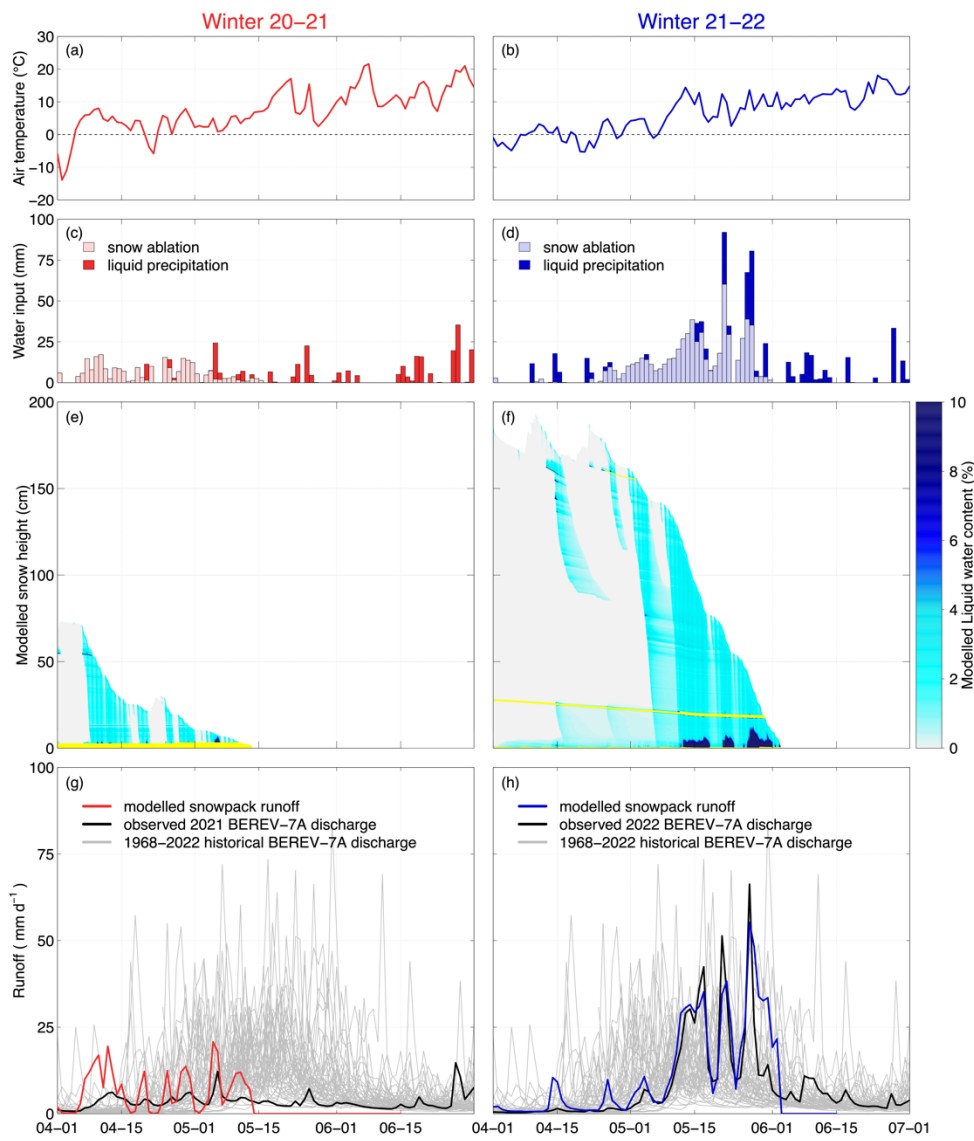

**Figure 11: Air temperature (a–b), daily difference in SWE, liquid precipitation (c–d), modeled snowpack liquid water content (e–f), simulated snowpack runoff and observed streamflow discharge (g–h) for winter 20–21 (left) and 21–22 (right). Air temperature and liquid precipitation are measured at the NEIGE site, located some 4 km north of the main study site and about 200 m lower in elevation. SWE is averaged for the canopy (75%), small gap (12.5%) and medium gap (12.5%) stations for representativeness of the study catchment. SNOWPACK simulations are representative of the subcanopy snowpack only. The streamflow discharge is monitored at the outlet of the BEREV-7A catchment. Ice layers are shown in yellow in (e) and (f), while the gray lines in (g) and (h) show historical measurements over the 1968–2022 period.**

## 4 Discussion

So far, our observations show that the low–snow and warm winter of 2020–2021 led to a slower melt, colder ground, enhanced
gradient metamorphism and ultimately to a reduced and less intense spring freshet than the reference winter of 2021–2022.
We used the SNOWPACK model to support our observational results under the canopy regarding the formation of a basal ice
layer, downward liquid water transport in the snowpack, and the resulting runoff in spring. Note that there are important
nuances to consider with respect to forest gaps and subcanopy snowpacks. Figure 12 provides a conceptual summary of our
results.

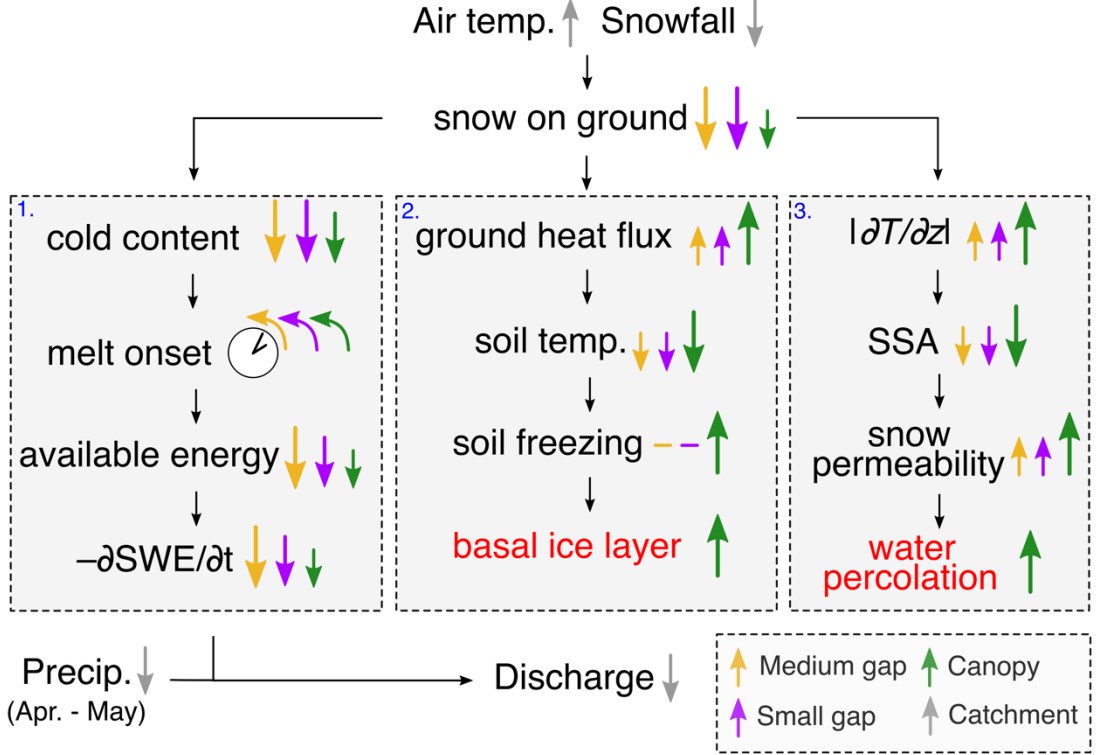

**Figure 12: Summary of the results. Upward arrows correspond to an increase and downward arrows to a decrease in the low–snow and warm winter with respect to the reference winter. The clock with counterclockwise arrows means that the process happens earlier. Results obtained from SNOWPACK simulations are in red. The yellow, purple, and green arrows indicate the effects in the medium gap, the small gap and under the canopy, respectively. The size of the arrows indicates the magnitude of the process at one location relative to the others. Large gray arrows indicate an analysis made for the entire catchment. Small black arrows show the causal link between the observations processes. Gray boxes refer to processes treated in this study (1. snowmelt dynamics; 2. soil thermal regime; 3. snow metamorphism).**

### 4.1 Low–snow and warm winter

In eastern Canada, as in other high–latitude and high–elevation regions, the snow cover extent is expected to decrease due to
warmer winter temperatures (Guay et al., 2015; Pepin et al., 2015; Kunkel et al., 2016). This region is also expected to receive
more winter precipitation in the future (Guay et al., 2015; Ouranos and MELCCFP, 2022). Therefore, the exceptionally warm

and dry conditions observed at MF in W2020-21 are not entirely consistent with the median climate projections for eastern Canada. However, these conditions did result in a snowpack that melted out 23 days earlier and in a maximum snow height that was 36% lower than the 1982–2022 reference period (Table 4). Based on these low snow accumulation conditions, the winter of 2020–21 is representative of what can be expected in eastern Canada with climate change even though the expected more abundant liquid precipitation may lead to snowpack modifications more significant than observed in W20–21.

Another feature of global warming in eastern Canada is the projected increase in the occurrence and intensity of ROS events (Il Jeong and Sushama, 2018), which has already been observed in other snow–dominated regions of the Northern Hemisphere (McCabe et al., 2007; Pall et al., 2019; Hotovy et al., 2023). The two ROS events observed at the beginning of 2020–21 were intense, with 44 and 106 mm of liquid precipitation in less than 36 hours each time. These two events reduced the snow cover thickness (Fig. 3a–c–e) and caused a large increase in streamflow discharge. However, overall, we observed fewer ROS than in the following year (8 versus 13), due to earlier melt–out in 2020–21 and dryer conditions in spring 2021.

## 4.2 Earlier and slower melt

During the low–snow and warm winter, snowpack $Q_{CC}$ was lower than in the reference winter. This is partly explained by a thinner snowpack with lower density (Fig. 3 and Fig. 10b–f–j). Warmer upper snow layers in W20–21 also contributed to the lower $Q_{CC}$. However, the lower snow layers were cooler in W20–21, which reduced the difference in $Q_{CC}$ between the two years. Our observations are consistent with simulations by Jennings and Molotch (2020) for alpine and subalpine sites in the western US, where $Q_{CC}$ is expected to decrease with increasing winter temperatures. Interestingly, the subcanopy snowpack had a larger $Q_{CC}$ than in the forest gaps, despite being thinner and lighter. This would imply an enhanced outgoing heat flux under the canopy that would considerably cool down the snowpack compared to both gaps. However, as noted in Sect. 3.2, this explanation would contradict the warmer snow surface under the canopy in both years (Fig. 4). This behavior is expected as radiative cooling in forest is more likely to take place at locations of higher sky–view fraction, such as gaps. This suggests a cold bias in the monitored temperature profile time series under the canopy which lead to an overestimation of the cold content. Timelapse images taken at the canopy station indicate that the snowpack seemed to grow thicker under the ultrasonic sensor than at the thermistor array in both years, which could explain the discrepancy between Fig. 3e–f and Fig. 4.

Since the snowpack had a lower $Q_{CC}$ in W20–21 in both the gaps and below the canopy, less energy was required to bring it to the melting point so snowmelt started earlier, consistent with what is expected in warmer winters (Barnett et al., 2005; López-Moreno et al., 2013; Guay et al., 2015; Jennings and Molotch, 2020). An earlier melt onset implied that net radiation was lower in W20–21, explaining a lower melt rate than in the reference winter (Fig. 5), which is consistent with a lower melt rate in that year (Fig. 6). Although it was not measured in this study, sensible heat flux may have contributed to snowmelt in gaps and under the canopy. This should be addressed in future modeling studies despite challenges in simulating turbulent fluxes in discontinuous forests (Conway et al., 2018). The higher rate of melting in the gaps coincides with greater incoming solar radiation than under the canopy. In fact, solar radiation is known to be the main driver of melting in these environments (Malle et al., 2019; Lawler and Link, 2011; Ellis et al., 2011). However, with less snow accumulating under the canopy due to

interception, the snowpack melted–out earlier than in both gaps, in both years. Our results support conclusions from previous studies as canopy interception exerts the main control on snow accumulation patterns whereas distribution of shortwave radiation is the main driver of ablation patterns (Lundquist et al., 2013; Mazzotti et al., 2023b; Dharmadasa et al., 2023). Overall, our results show that snowmelt dynamics are highly variable at the local scale in forests due to the discontinuous canopy structure. It underscores the importance of using high–resolution canopy structure mapping in snow models to accurately predict snowmelt in forests.

### 4.3 Increased frozen soil under the canopy

A thinner snowpack in the low–snow winter allows more heat loss from the ground to the atmosphere, and as such, a larger ground heat flux, which led to a cooler soil than for the reference winter (Fig. 7; Table 5). Under the canopy, this phenomenon was intensified as indicated by subzero temperatures in the top 5 cm of soil in both years. These observations are supported by a modeled basal ice layer in both years, the formation of which is favored by a frozen soil (Albert and Hardy, 1993; Westermann et al., 2011). The thicker simulated ice layer in W20–21 is consistent with the enhanced soil freezing observed in that year. Overall, our results suggest that the heat loss was sufficient to favor soil freezing under the canopy, but not in the forest gaps. The topsoil thermal conductivity was measured at 0.8 W m$^{-1}$ K$^{-1}$ in the summer 2021 at the canopy station (Fig. S2). Given that the temperature difference between depths of 5 and 20 cm in the soil varies between 0.8 and 1.2 °C, we can readily estimate from Fourier's law applied to the top 5 cm of soil (Eq. 5) that the average ground heat flux is 5.3 W m$^{-2}$. This is lower than the estimated snow heat flux under the canopy (Fig. 7e–f) which explains why the topmost subcanopy soil layers froze in both years. These findings are in partial agreement with Stadler et al. (1996), who observed a cooler ground under the canopy, where less snow accumulates. However, the authors also observed ground freezing in a nearby gap. This could be explained by the much lower snow accumulation at their study site compared to what was observed in the MF in both years. Slater et al. (2017) assessed the influence of snow depth on the season-average temperature difference between the air and the soil. Based on data from numerous northern, arctic, and alpine sites, they concluded that beyond a mean snow depth of about 20 cm, air and soil temperature were decoupled. Although that study did not include sites from eastern Canada, we chose to compare our results with the observations presented in Fig. 3 from their paper. Our mean snow depth, for both years and below canopy and in gaps, were all above 20 cm, and we indeed found very small variations in air-soil temperature differences. This supports our previous findings that the exceptional low-snow and warm conditions met in winter 20–21 had almost no effect on the thermal insulation between the soil and the air due to thickness and properties of the snow (Fig. 7; Table 5). The increased snow faceting in W20–21 may also have contributed to a more efficient insulation of the ground despite a thinner snow cover. Fig. 8 suggests that the soil has a similar pore space under the canopy and inside gaps, as it saturates at a volumetric water content (VWC) between 0.50 and 0.55 at all monitored sites. However, the seasonal low of VWC is at 0.38, 0.42 and 0.48 in the medium gap, the small gap and under the canopy, respectively, suggesting a higher water retention under the canopy and therefore smaller pores than in the gaps. This implies less potential for water content to increase in response to snowmelt or ROS event. The slower increase in VWC under the canopy at the onset of the 2021 snowmelt (Fig. 8a) also suggests that

infiltration was limited, but not completely restricted compared to what was observed in the gaps. Based on previous experiments in the Montmorency Forest, Proulx and Stein (1997) concluded that ROS greater than 20 mm followed by a cold spell would cause macropores in the boreal soil to become ice–filled, limiting subsequent infiltration. According to this criterion, the two ROS of December 2020 (42 and 105 mm) should have favored the blockage of soil pores in W20–21. Overall, our results are in accordance with Demand et al. (2019), who observed a reduced infiltration into frozen soil compared with unfrozen but highly saturated soil during sprinkling experiments over a sandy loam in southern Germany.

### 4.4 Larger temperature gradients and snow permeability

Our observations show that a thinner snowpack in W20–21 reduced load compaction and therefore limited snow densification, compared to W21–22. A thinner snowpack also led to higher vertical temperature gradients ($|\partial T/\partial z|$) in the low–snow and warm winter (Fig. 9), resulting in more pronounced faceting, lower SSA, and a higher permeability ($K_s$) than in W21–22 (Fig. 10). SNOWPACK simulations suggest that the enhanced gradient metamorphism in W20–21 resulted in a slightly faster water percolation in the snow cover. In terms of SSA, no significant difference was observed between the two years for the lowermost part of the snowpack, where we observed DH. This is because $|\partial T/\partial z|$ in both years was sufficient to favor DH development, and the SSA of DH remained always around $10 \pm 2$ m$^2$ kg$^{-1}$, regardless of its stage of development (Bouchard et al., 2022; Domine et al., 2018). In February and March, the $|\partial T/\partial z|$ was larger in W20–21 than for the same period the next year. This resulted to a faster development of FC and DH and to a lower SSA in the middle part of the snowpack. Since snow permeability is closely related to snow metamorphism and $|\partial T/\partial z|$ (Domine et al., 2013; Taillandier et al., 2007;Calonne et al., 2012), $K_s$ shows a similar pattern to SSA in the snow profiles. When comparing the snowpack under the canopy to the one in both gaps, we observe that temperature gradient metamorphism was enhanced under the canopy, in line with observations from Bouchard et al. (2022). Interestingly, our field measurements in forest gaps and under a boreal canopy contradict Domine et al. (2007) who, based on general considerations, suggested a decrease in snow permeability with global warming mainly due to increased snow surface temperature. Our results suggest that if warmer winters lead to a large decrease in snowpack thickness, this could override the increase in surface temperature and lead to higher $|\partial T/\partial z|$ and permeability.

### 4.5 Meteorological conditions versus forest structure

A larger difference in $H_{s,max}$ between the two years than between the sites suggests that weather conditions had a greater effect on the snow accumulation than forest structure. Lower snow accumulation at the onset of snowmelt in the warm year compensated for the much slower melt than in the reference year, so the difference in snowmelt duration between both years was less pronounced than the difference between the sites. Since weather conditions also controlled the onset of the melt period, as indicated by snowmelt starting on the same day at all sites in each year, we observed a melt-out date difference that was much greater between years than between sites. Overall, this suggests that the meteorological forcing was more important in influencing snow accumulation and melt than the canopy structure.

One might also expect that the difference in soil temperature between years would be more pronounced than between sites, as a consequence of a greater difference in snow height. However, our results show the opposite. In fact, the evolution of the soil temperature followed a similar pattern in both years, with a slightly lower temperature in the warm year for a given depth. In contrast, the differences in soil temperature between gap and canopy sites were much more pronounced, in particular near the ground surface, with freezing observed only under the trees. This suggests that the forest structure had a greater influence on the soil thermal regime than the weather conditions at this study site during these two years.

A larger difference in $|\partial T/\partial z|$ between gaps and canopy sites than between W20–21 and W21–22 from November to January indicates that the canopy structure had a strong influence on the $|\partial T/\partial z|$. However, the difference in $|\partial T/\partial z|$ between both years became more pronounced in February and March, which also suggests that the relative influence of weather conditions on the temperature gradient increases during winter. Since the differences in snow density, SSA and $K_s$ between gaps and canopy and between seasons were all significant, we cannot distinguish which factor (meteorological conditions and forest structure) dominates regarding the evolution of snow properties.

## 4.6 Reduced spring streamflow

Figure 11 clearly shows that the spring freshet was earlier and reduced in the low–snow and warm winter compared to the reference winter. Both modeling and observational results support that an earlier and slower melt exerted a major influence on streamflow regime, which is consistent with the work of Musselman et al. (2017) in the western United States. A thinner snowpack in the warm year mainly explains why the wetting front in the SNOWPACK simulations reached the ground much faster than in the reference year. The modeled percolation rate was slightly higher in W20–21 than in W21–22, but this difference remains small compared to the difference in snowmelt timing between the two years. Also, the observed and simulated thicker basal ice formation in W20–21 did not result in a faster hydrological response at the outlet of the catchment, as might be expected.

The large difference in spring runoff between the two years can further be attributed to lower precipitation as ROS in the spring of the warm year. Indeed, ROS accumulation was nearly four times lower in W20–21 than in W21–22. This can be attributed to dryer conditions in spring (Fig. 2a) and also to a short–lived snowpack, limiting the exposure of the snowpack to rainfall in spring (Cohen et al., 2015). The delayed and lower discharge response to modeled snowpack runoff in W20-21 suggests that some of the meltwater and liquid precipitation infiltrated into an unsaturated soil and recharged the aquifer (Schilling et al., 2021). It is likely that this also happened in April of 2022, before snowmelt (Fig. 11h). In contrast, the higher discharge, which was also synchronized with snowpack runoff simulations, suggest that the soil was saturated and subsurface flow contributed to a greater streamflow discharge during the 2022 snowmelt. These contrasting results are not surprising given that the spring daily water input was often much higher in 2022 than in 2021. Overall, in light of our observations and SNOWPACK simulations, it appears that the effects of both soil freezing and snow structure on the timing and amplitude of runoff are of secondary importance compared to the influence of an earlier and slower snowmelt and spring liquid precipitation.

## 4.7 Limitations and shortcomings

In this study, we examined snow accumulation and melt dynamics from highly detailed in situ measurements. However, our experimental setup lacked lysimetric measurements to quantify the effect of soil freezing and snowpack permeability on runoff. Although there are multiple challenges related to lysimeters (Kattelmann, 2000; Floyd and Weiler, 2008), we recommend using those in future studies for quantitative assessment of infiltration. In the absence of lysimetric measurements, soil moisture monitoring can provide information on the occurrence of infiltration. Therefore, we recommend monitoring soil VWC at multiple forest gaps and canopy locations in future studies to better understand soil infiltration dynamics in discontinuous boreal forests under a warming climate. Furthermore, a complete interpretation of soil liquid water content data would benefit from the knowledge of soil granulometry and hydraulic conductivity.

Although this observational and modeling study combines snow accumulation and melt dynamics, soil freezing and snow microstructure in one unique dataset representative of the humid boreal forest, the analysis itself remains a case study. Other specific characteristics of the study site such as the slope, the aspect and the surrounding topography, influence the formation and the ablation of the snowpack in forested environments (Lundquist and Flint, 2006; Ellis et al., 2013; Mazzotti et al., 2023b). To assess the impact of low–snow and warm winter conditions on snowpack dynamics at broad scales, the impact of these factors should be considered.

Although the SNOWPACK model allowed for a more thorough interpretation of the hydrological influence of snow properties, the simulations were limited to subcanopy locations. Coupling a multilayer snow model, such as SNOWPACK or Crocus (Vionnet et al., 2012), with a detailed representation of canopy structure, such as that found in FSM2 (Mazzotti et al., 2020), would be required to properly simulate snowpack evolution in forest gaps. Recent work by Mazzotti et al. (2023a) is promising in this regard.

## 4.8 Climatic, hydrological and ecological implications

In the low–snow and warm winter, snowmelt started much earlier, but occurred over a longer period at all sites, resulting in an earlier melt–out compared to the reference winter. A shorter–lived snowpack in the boreal forest has climatic impacts, as the net shortwave radiation increases due to a reduction in surface albedo (Manninen and Stenberg, 2009) and contributes to a positive feedback loop that enhances global warming (Thackeray and Fletcher, 2015). Moreover, a decrease in snow cover extent in the boreal forest may increase the risk of summer hydrological drought due to a lower groundwater recharge (Van Loon et al., 2015). Based on the definition of Van Loon et al. (2015), the conditions in spring 2021 (Fig. 11) have the characteristics of a snowmelt drought.

With increasing warming–induced ROS events, the formation of melt–freeze layers, as we observed at the base of the W20–21 snowpack, and within the snowpack are likely to become more common. These could alter liquid water pathways, favoring water ponding and lateral flow over percolation (Eiriksson et al., 2013; Paquotte and Baraer, 2022; Webb et al., 2018). In addition to altering downward liquid water transport through the snowpack, melt–freeze formations can limit soil–atmosphere

gas exchange, promoting hypoxic conditions in subnivean environments (Crawford and Braendle, 1996). Melt–freeze formations also limit access to food and movement of subnivean mammals (Johnsen et al., 2017; Poirier et al., 2019), and

605 restrict foraging by large herbivores (Hansen et al., 2011; Schmelzer et al., 2020). As we observed in December 2020, intense ROS events also trigger winter snowmelt. This promotes low–snow conditions that can intensify soil freezing, as observed under the canopy snowpack in W20–21. Deeper soil freezing in forests decreases microbial activity and soil respiration in winter, thus reducing soil nitrogen recycling (Patel et al., 2018; Yang et al., 2019). In turn, this favors carbon accumulation which offsets increased soil respiration in summer (Patel et al., 2018). In summary, it seems clear that changes in snowpack

thickness and structure with climate warming have both hydrological and ecological implications in the boreal forest.

**5 Conclusion**

Using dedicated field observations, along with SNOWPACK simulations, we investigated the effects of a low–snow and warm winter on snow accumulation and melt dynamics, on soil thermal regime and moisture, and on snowpack physical properties under the canopy, and inside two forest gaps in a humid boreal site of eastern Canada. More precisely, we focused on winter

2020–21 (W20–21), which was exceptionally warm and dry, comparing it with a reference winter (W21–22), closer to climate normals. The experimental setup included snowpack and the soil thermal regime monitoring, along with monthly snow pit observations. Our results show that the snowpack was generally half as thick and started to melt earlier in the low–snow and warm winter, compared to the reference winter. This increased soil freezing under the canopy, but not in forest gaps, where the soil remained unfrozen, as for the reference winter. Although the snow surface was warmer in W20–21, the thinner

snowpack led to an increase in the vertical temperature gradient, so kinetic growth was enhanced. This resulted in a higher snow permeability during the warm, low–snow year, particularly under the canopy. Although enhanced soil freezing and larger snow permeability, supported by simulations of a thicker basal ice layer and faster percolation, point toward faster runoff and larger peak flow, our results suggest that these are of second importance as low snow accumulation in winter, early snowmelt and low spring rainfall in spring led to a significantly lower spring freshet in the warm year.

The conditions experienced in the winter 2020–21 at Montmorency Forest, such as warmer air, less snowfall, and a thinner snowpack, were exceptional compared to the past climatology. However, these conditions are likely to become more frequent in eastern Canada with climate change. Although this work is limited to a two–year comparison within a small catchment, it highlights the many potential effects, all together, of a changing climate on snow hydrology in a discontinuous boreal forest through a unique set of highly detailed process–level observations. These are highly valuable for the snow science community

as they will help improve existing modeling tools and develop new ones to address future challenges in snow hydrology.

*Code and data availability.* Documented code of SNOWPACK version 3.6.0 is available on GitLab (*https://gitlabext.wsl.ch/snow-models/snowpack*). Data from snow pit measurements and from the monitoring stations in the medium and small gaps, and under the canopy are freely available at https://doi.org/10.5281/zenodo.8213204.

*Author contributions.* BB, DFN and FD designed the study. BB and ET collected and treated field data. BB ran the simulations and conducted the analysis of the results with inputs from DFN, FD, FA, and TJ. BB wrote the manuscript, with feedback from all authors.

*Competing interests.* The authors declare that they have no conflict of interest.

*Acknowledgements.* The authors thank the staff of the Montmorency Forest for helping us logistically with field visits. We also thank Charles Villeneuve and Kino Leroux for preparing snowmobiles trails before our visits. The authors also thank Éric Boucher, Christian Juneau and Antoine Thiboult for helping in the deployment and the maintain of the monitoring stations. We thank Pierre-Erik Isabelle for providing radiation dataset. We finally thank all the people that accompanied Benjamin Bouchard and Etienne Tremblay on the field, especially the members of PÉGEAUX and the graduate students, post–doc, and research associates from the Hydrometeorology Lab at Laval University. We would finally like to thank two anonymous reviewers for their valuable comments and suggestions which helped improve the quality of the manuscript. The work of BB was founded by the Natural Sciences and Engineering Research Council (NSERC) and Sentinel North program. The authors acknowledge the financial contribution of Environment and Climate Change Canada through the Grants & Contributions program (projects #GCXE20M016 and #GCXE22M013) and the Cold–region climate projection for hydrological applications (EVAP-2; project #ALLRP 549108 – 19).

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
