# Peer review of "How does a warm and low–snow winter impact the snow cover dynamics in a humid and discontinuous boreal forest? Insights from observations and modeling in eastern Canada"

_Hydrology and Earth System Sciences, 2023_

## Author Comment (AC1)

**Responses to the Anonymous Referee #1**
* * *
First of all, we wish to thank anonymous reviewer #1 for providing constructive and insightful comments and suggestions. Based on these, we plan significant changes to the original manuscript. We hope that these changes, if accepted, will make the paper clearer and easier to read. In particular, we have carefully revised the Introduction and the Discussion sections. Our answers below are in blue, whereas excerpts from the manuscript are in blue *italics* with modifications in **bold**.
* * *
This paper investigates the impact of a dry and warm winter on the snow dynamics in a discontinuous boreal forest in northeast Canada. Comparing observations of snow dynamics in a low-snow winter with a winter close to normal conditions can give insights about expected future changes. In boreal forests, the snow dynamics differ between open gaps and under the canopy. The study uses observations at three nearby locations (under the canopy, small gap, large gap) in a small catchment in northeastern Canada. Measurements of snow physical properties, its thermal regime, and soil measurements were taken over two consecutive winters, which represent low-snow and normal conditions. Results show less snow accumulation and an earlier melt, which was slower due to lower radiative forcing, in the low-snow year. The topsoil layers were cooler and under the canopy soil freezing was enhanced in the warmer year. The spring freshet occurred earlier but was less intense, due to slower melt.

Generally, this is a well-written paper, which presents a lot of interesting observational data on various aspects of snow dynamics accompanied by relevant figures. The paper is well structured. However, some discussion about the limitations of the approach to give insights into future changes are missing and the second objective is not addressed in-depth.

**Specific Comments: Major**

1. **Exceptionally dry year**

a) At your study site, the winter 20/21 was exceptionally warm AND dry. You mention that it is "plausibly representative of future winters" (l.23). In the introduction you described the expected warming levels in boreal forests. However, I missed an introduction to how future precipitation is projected to change for boreal forests and eastern Canada. From the statement in l. 39 I assume, that annual precipitation is projected to increase. Is winter precipitation also expected to increase?

This is a very good point. Projections of future precipitation in the boreal forest of eastern Canada are spatially variable, although winter precipitation is generally expected to increase. However, it is expected that solid precipitation will increase in the north and decrease in the south (Guay et al., 2015) [doi:10.1080/07011784.2015.1043583], as for our study site. The interannual variability of precipitation and temperature is also expected to increase in future (MELCC and Ouranos, 2022) [https://cehq.gouv.qc.ca/atlas-hydroclimatique/], making warm and dry winters more likely. We will rephrase the manuscript with the following changes to make this clearer (please also see comment 7):

**l. 38 to l. 41 (Introduction):** "***Projections*** *for the boreal forest of eastern Canada**, characterized by humid and cold conditions in winter (D'orangeville et al., 2015; Isabelle et al., 2020),** point towards an increase in winter streamflow and an earlier spring freshet **with more snow accumulation in the north***

*and less in the south (Guay et al., 2015). The interannual variability of precipitation and temperature is also projected to increase, making warm and dry winters more likely (MELCC, Ouranos 2022).*"

Note that we also propose to remove the reference to Valencia Giraldo et al., (2023) [doi:10.3390/w15030584] on l. 40 as we realized that this study did not cover boreal catchments. We would also remove the reference to Shook and Pomeroy (2012) [doi:10.1002/hyp.9383] on l. 38, as it relates more to the Canadian prairies than the boreal forests.

b)  If future winter precipitation is projected to increase, the winter 20/21 is not representative of the projected future. I would have expected a discussion on this and how this impacts the conclusions you can draw from your observational study for future changes in snow dynamics and runoff. How do your results differ from what you would expect with climate change?

It is exact that a low winter precipitation trajectory in the future contrasts with average climate projections for eastern Canada. However, greater variability is also forecasted, so that low precipitation is expected to be fairly common. Furthermore, the low-snow conditions of winter 20-21 (low snow accumulation and early melt-out) are representative of future winters for the southern region of the boreal forest of eastern Canada. The conditions of winter 20-21 are also representative of extremes that may become more common in the future as interannual variability increases (IPCC, 2022) [doi: doi:10.1017/9781009325844]. In any case, a single winter can only be representative of a fraction of future winters. We suggest the following changes to the manuscript:

**l. 14-15 (Abstract):** "*In the boreal forest **of eastern Canada**, winter temperatures are projected to increase substantially by 2100. **Although this region is also expecting more precipitation, an increase in the interannual variability of precipitation and temperature is making warm and low-snow winters more likely in the future** resulting in a reduction in snow cover thickness and duration.*"

**l. 23 (Abstract):** "*…an exceptionally low-snow and warm winter, **projected to occur more frequently in the future**, and…*"

**l. 393 to l. 399 (Discussion):** "*In eastern Canada, as in **other** high-latitude and high-elevation regions**, the snow cover extent is expected to decrease due to warmer winter temperatures (Guay et al., 2015; Pepin et al., 2015; Kunkel et al., 2016). Although precipitation is expected to increase for this region, an increase in the interannual variability of precipitation and temperature means that warm and low-snow winters are also projected to become more frequent (Ouranos and MELCCFP, 2022).** Winter 2020–21 at MF received 211 mm less solid precipitation, was warmer by 4.6°C in DJF, and snow melt-out occurred 34 days earlier than in 2021–22, which is more representative of the prevailing local climatic conditions **despite a slight negative anomaly in DJF temperature** (Fig. 2; Table 4). Indeed, W20–21 was exceptional in that it had both the lowest snowfall in the last 40 years and was one of the warmest winters in that period (ranked fourth).*"

**l. 526 to l. 528 (Conclusion):** "*The conditions experienced in the winter 2020–21 at Montmorency Forest, such as warmer air, less snowfall, and a thinner snowpack, were exceptional compared to the past climatology. **However, these conditions are likely to** become **more frequent** in eastern Canada with climate change.*"

c)  In l. 54 you state that more frequent and intense winter rainfalls are expected with climate change. Such increased winter rainfall could lead to more rain on snow events, especially at the beginning and end of the winter, which likely influences the discharge. Could you please elaborate on the above aspects in your discussion?

Based on comment 3b, we propose to rearrange the paragraph at l. 42 to l. 55, and in doing so the statement at l. 54 would be removed. We observed two intense rainfall events on 1 and 24 December 2020, causing a major increase in streamflow discharge (Figure R1.1). However, in part because the snow melted out early in spring 2021, we observed fewer ROS events than the following year. It has been shown that for cold regions, warmer temperatures will favor the occurrence of ROS, whereas the reduction in snow cover longevity will have the opposite effect (Cohen et al., 2015) [doi:10.1002/2015GL065320]. We suggest modifying section 4.1 of the manuscript as follow:

**l. 399 to l. 405 (Discussion):** "*Another feature of global warming in eastern Canada is **the projected increase in the occurrence and intensity of ROS events (Il Jeong and Sushama, 2018), which has already been observed in other snow-dominated regions of the Northern Hemisphere (McCabe et al., 2007; Pall et al., 2019; Hotovy et al., 2023)**. **The two ROS events observed at the beginning of W20-21 were intense***, with 44 and 106 mm of liquid precipitation **in less than 36 hours** each **time.** These two events reduced the snow cover thickness (Fig. 3a–c–e) **and caused a large increase in streamflow discharge. However, overall, we observed fewer ROS than in the following year (8 versus 13), due to earlier melt-out in W20-21 and dryer conditions in spring 2021.**".*

[Figure]

Figure R1.1: Discharge at the DEH station 051004 from October to July 2020-21 and 2021-22. The black arrows indicate the runoff from the ROS events of 1 and 24 December 2020.

**2. Evaluation of the effect of snow dynamics on spring runoff**

In the abstract you mention a research gap: "Although the effects of warmer winters on snow-related processes are well documented, their interactions to influence the spring runoff in evergreen forest remain poorly understood." (l.17-19). It sounds like this is one of the two research gaps you would like to address in your study, which is made clear in the introduction: "The second one is to evaluate how these factors interact together to modulate spring runoff." (l.71). From the sentence in the abstract and the objective I would expect that you look at the interactions of several processes to distinguish their individual influence on spring runoff. However, in the results, you show discharge measurements only in relation to air temperature and SWE changes. You do not consider, how individual factors influence the SWE changes and the discharge, such as the increase in snow permeability and the soil freezing. You do mention that the decrease in available energy in the melt period probably decreases the magnitude of the spring freshet, but the relationship between, e.g. the effect of the earlier onset of the melt season in relation to the infiltration vs. surface runoff of snow melt remains unclear. To estimate the effect of increased snow permeability and soil freezing on the spring freshet, you would need information about the partitioning between infiltration and the surface runoff.

a) I would expect a more in-depth discussion about the limitations of achieving objective 2, e.g. why you did not measure infiltration and surface runoff.

We suggest dropping the second objective (see comment 2c). Infiltration and surface runoff were not measured because the instrumental setup was lacking lysimetric measurements. It would have been possible to estimate infiltration and runoff by using a snow model but this is beyond the scope of this observational study. Overall, we agree that these limitations should be mentioned in the manuscript. Therefore, we would rearrange entirely section 4.6 as follows:

**l. 478 to l. 492 (Discussion)**: "*In this study, we examined snow accumulation and melt dynamics from highly detailed in situ measurements. **However, our experimental setup lacked lysimetric measurements to quantify the effect of soil freezing and snowpack permeability on runoff. Although there are multiple challenges related to lysimeters (Kattelman, 2000; Floyd and Weiler 2008), we recommend using those in future studies for quantitative assessment of infiltration. In the absence of lysimetric measurements, soil moisture monitoring can provide information on the occurrence of infiltration. Therefore, we recommend monitoring** soil VWC at multiple forest gaps and canopy locations in future studies to better understand soil infiltration dynamics in discontinuous boreal forests under a warming climate. Furthermore, a complete interpretation of soil liquid water content data would benefit from the knowledge of soil granulometry **and hydraulic conductivity**.*"

*Although this observational study combines snow accumulation, melt dynamics, soil freezing and snow microstructure in one unique dataset representative of the humid boreal forest, the analysis itself remains a case study. Other specific characteristics of the study site such as the slope, the aspect and the surrounding topography, influence the formation and the ablation of the snowpack in forested environments (Lundquist and Flint, 2006; Ellis et al., 2013; Mazzotti et al., 2023). To assess the impact of low-snow and warm winter conditions on snowpack dynamics at broad scales, the impact of these factors should be considered.*

***As the conclusions of this study are based solely on observations, it would be interesting to pursue analyses with models simulating water and energy exchanges along the atmosphere-forest-snow-soil continuum. Multilayer snow models such as SNOWPACK (Bartelt and Lehning, 2002) or Crocus (Vionnet et al., 2012) would be adequate tools in this regard, as they are able to estimate water transport in a one-dimensional snow column and the infiltration into the soil.***"

b) Moreover, regarding my first comment on the exceptionally dry winter, I would expect a discussion about how spring runoff is affected if winter precipitation increases and rain on snow events increase.

We agree with the comment. First, we suggest expressing the rainfall contribution as rainfall in the presence of a snow cover in spring instead of liquid precipitation in April and May. Second, we would discuss the role of other factors (soil freezing and snow structure) on spring runoff. The following changes are suggested:

**l. 364 (Results)**: "*In April and May, **rain-on-snow accounted for 54 mm in 2021 compared to 202 mm the following year**.*"

**l. 470 to l. 476 (Discussion)**: "***In the warm winter, ROS accumulation was nearly four times lower than in the reference winter. This can be attributed to dryer conditions in spring (Fig. 2a) and also to a short-lived snowpack, limiting the exposure of the snowpack to rainfall in spring (Cohen et al., 2015).*** *Given a temporal lag of 3 to 5 years between the recharge of the aquifer and groundwater outflow at BEREV-7A (Schilling et al., 2021), **streamflow only increases from rainfall or snowmelt. Although***

*enhanced soil freezing and higher snow permeability favor larger runoff, we cannot quantify these contributions with the available data. However, in light of our results, it appears that both soil freezing and snow structure are of secondary influence on spring streamflow while snowmelt dynamics and precipitation are the main drivers. Sub-daily streamflow measurements would be needed to assess how ground freezing and snow permeability affect the time lag between small ROS and melting events and the hydrological response of the catchment.*"

      c) The second objective in general is addressed in much less detail in your study than the first one, which is addressed extensively. For example, in the methodology, it is not clearly introduced which methods are used to achieve objective 2. Also, the title of the paper only encompasses the first objective. Therefore, I suggest regarding the evaluation of the effect of snow dynamics on spring runoff not as a second objective, but rather as a further analysis and frame the paper accordingly.

This is a good remark. We agree that we do not have a sufficient dataset to quantitatively meet the second objective. Moreover, a daily timestep for streamflow measurements are too coarse to assess how ground freezing and snow permeability affects the time lag in discharge response of ROS and melting. Therefore, we suggest removing the second objective from the study. We suggest deleting the sentence from l.17 to l.19 in the abstract and modifying the Introduction and the Discussion accordingly (please see comment 3b).

**3. Introduction: clearly identify the research gap**

      In general, I think the paper is well-written and well-structured. However, I struggle a bit with the introduction, which could be more concise and better structured, I think.

      a) The introduction about soil freezing is very long in comparison to the introduction of other background information and processes.

We agree that the paragraph on soil freezing in the introduction is too long and that the research gap is not clearly expressed. We looked more thoroughly at the literature and found that forest cover as well as low-snow conditions favor low snow accumulation, ground freezing and infiltration. However, the impact of both factors on ground thermal regime have not been investigated together at the plot scale. We believe that our results contribute to filling this research gap. Here is the modified and shortened version of the paragraph that we propose:

**l. 42 to l. 55 (Introduction):** "*It has been shown that the ground thermal regime is strongly influenced by the amount of snow accumulation (Zhang et al., 2005; Slater et al., 2017). In forests, the spatial pattern of soil temperatures is difficult to determine because snow depth is highly variable (Mellander et al., 2005). Observations from a subalpine forest plot in Switzerland show that frost penetrates the ground deeper under tree crowns due to less snow accumulation than in forest gaps which reduces the infiltration and increases surface runoff (Stadler et al., 1996). Infiltration is also limited during low-snow winters due to a thinner snowpack that favors soil freezing (Hardy and al., 2001; Shanley and Chalmers, 1999). It is clear that both canopy structure and snow conditions influence the ground thermal regime, soil freezing and infiltration. However, it is not well understood which of these two factors predominates over the other because they have not been investigated simultaneously in a single study.*"

      b) I also had difficulties in identifying the exact research gap you would like to address based on the introduction. It remains unclear to me whether changes in snow dynamics in relation to warmer winters are known in boreal forests or in other biomes in general. The statement in

line 39-40 contradicts the statement in l. 18 ("effects of warmer winters on snow-related processes are well documented"). This makes it difficult for the reader to understand what similar relevant research has been done and to identify the existing research gaps you aim to address with this study.

The main research gap that motivates our work is that winter weather conditions and canopy structure have not been studied together to see how they influence snowmelt dynamics, the ground thermal regime, and the physical properties of the snowpack. We suggest the following modification to the manuscript:

**l. 69 to l. 76 (Introduction):** "***The main research gap that motivates our work is that winter weather conditions and canopy structure have not been studied together to see how they influence snowmelt dynamics, the ground thermal regime, and the physical properties of the snowpack. Thus, the objective of this study*** *is to quantify the effect of a low-snow and warm winter* ***on the aforementioned processes*** *in a humid and discontinuous boreal forest. To assess this, we compared snow melt, snow physical properties, soil freezing, and spring runoff at a small catchment* ***in the south part of the*** *boreal forest of eastern Canada, for two consecutive winters. One winter was exceptionally warm and dry, while the other was colder, with precipitation amounts similar to the standard climatology of the study region. These contrasted conditions represent an ideal comparison to investigate some expected effects of climate change. Extensive snow monitoring and pit measurements were conducted to* ***achieve the research objective****.*"

The last part of the introduction is very well written (l.69-82).

Thanks!

4. **Snow Stratigraphy results:**

   a) l. 335: "At all sites, there was a greater proportion of faceted crystals (FC) and depth hoar (DH) in the snowpack during the low-snow year than during the reference year". However, I see from Figure 10 that the portion of light blue and dark blue colour (FC and DH) is smaller in the low-snow year than the reference year for the canopy.

This is explained by the thicker basal layer of melt-freeze polycrystals (MFpc) in W20-21. This MFpc layer was initially faceted crystals (FC) and was exposed to gradient metamorphism throughout the rest of the winter. Overall, the sum of MFpc, FC and depth hoar (DH) is higher in winter 20-21 than in winter 21-22 at all locations. We propose to modify the manuscript accordingly (please see comment 4c for the suggested changes).

   b) l. 336: "In contrast, we observed fewer rounded grains (RG) in W20–21 than in W21–22." This does not seem to be the case for the canopy looking at Figure 10.

Indeed, this only applies to the gaps. Suggested changes are presented in the response of comment 4c.

   c) l. 336-337: "In both years, FC and DH layers were proportionally thicker under the canopy than in the gaps." From Figure 10, I see that the light blue color (FC) covers a smaller proportion of the snow height in the canopy than in the gaps, which contradicts your statement. Maybe you mean that FC and DH layers combined were proportionally thicker?

That is indeed what we meant. We will make this clearer in the revised manuscript. Here are the changes that we propose for the third paragraph of section 3.4:

**l. 335 to l. 341 (Results):** "*At all sites, **the top of the faceted crystals layer (FC) rises higher in the snowpack in the warm and low-snow year than during the reference year. In winter 20-21, this height includes FC and depth hoar (DH), as well as a thick layer of melt-freeze polycrystals (MFpc) resulting from the December 2020 ROS. In this basal layer, we also observed** FC and DH but these are secondary to clusters of polycrystals. As a result, we observed* fewer rounded grains (RG) in W20–21 than in W21–22. In both years, **the combination of** FC and DH layers **was** proportionally thicker under the canopy than in the gaps and the thickness of the DH was noticeably higher under the canopy. **Overall, the level of faceting is the greatest in the snowpack under the canopy during the warm and low-snow year.**"*

**Specific comments: Minor**

**5.** l. 27: warm year instead of warmest year

This will be corrected.

**6.** l. 27: "Overall, we observe that the spring streamflow discharge was significantly reduced in the warmest year due to a slower melt and low precipitation in April and May." I would argue that it is mostly reduced due to less snow accumulation in winter and thus less snow melt that can contribute to the spring freshet. Why did you not mention this aspect?

We agree that this aspect should be mentioned. We suggest the following changes:

**l. 27 to l. 29 (Abstract):** "*Overall, we observe that the spring streamflow discharge was significantly reduced in the **warm** year due to **less snow accumulation**, a slower melt and low precipitation in April and May.*"

**7.** l. 37 "rather dry regions" and l. 39 "humid boreal forest": What is rather dry and humid in the context of a boreal forest? Can you provide a definition? Do you expect different behaviors?

We defined dry and humid boreal regions based on the water availability index map (see Trabucco et al. (2019) [doi:10.6084/m9.figshare.7707605.v3] and Figure 1a from D'Orangeville et al. (2015) [doi:10.1126/science.aaf4951]). We also characterized the boreal forest of eastern Canada as being humid according to Isabelle et al. (2020) [doi:10.1016/j.agrformet.2019.107813]. Based also on comment 1a and comment 9 from the referee #2, we suggest removing the sentence from l. 36 to l. 38.

**8.** l. 100: "The stations were located in the vicinity" Please include the distance to the flux tower

The stations were located within 30 m from the flux tower. A typo was also noted (the flux tower is 20 m high, not 15 m). We suggest that the manuscript be changed accordingly:

**l. 100 - l. 101 (Methods):** "*The stations were located **within 30 m** of a **20**-m flux tower*"

**9.** l. 130: lowest -> lower, highest -> higher, otherwise confusing it if is really only to probes

Thank you. We will do the change.

**10.** l. 166: the subscript should be "i", I think, but "l" is used

Thank you. This will be corrected.

**11.** l. 233, 234: It would be helpful for the reader if you can also give the precipitation anomaly in percent in comparison to long-term mean.

This is a very good suggestion. In winter 20-21 the anomaly was –53%, while the next year it was +1%. This will be added in the revised manuscript.

**12.** l. Line 265, 315: how do you define the onset of snowmelt?

In this work, we were interested in the beginning of the spring snowpack runoff. Therefore, the onset of snowmelt was defined as the increase in soil volumetric water content to a maximum in spring (see Figure 7). It occurred on 10 April 2021 in the first year and on 3 May 2022 in the second year. We suggest the following addition to the Methods section:

**l. 126 (Methods):** "*We also define the onset of snowmelt as the beginning of snowpack runoff, when the soil VWC reaches a maximum in spring.*"

**13.** l. 230, 255, 283, 293, 309, 321, 331, 369: You always use the same sentence structure: "Figure X shows …". These sentences basically repeat what can be seen in the figure caption. Stating the same in the main text is not necessary. To make the text more concise, I suggest removing these sentences and referring to the figures after the first statement about the results shown in the figure, e.g: W20–21 was the driest winter of the 1982–2022 period, with 199 mm recorded from January to April (JFMA), including 167 mm of solid precipitation (Fig.2, Table 4).

Thank you for this remark. We totally agree that this structure makes the manuscript heavier. We agree to remove the sentences at l. 230, 255, 283, 293, 321, 360 and refer to the figures accordingly in the text. The sentence at l. 309 is a statement about the figure so we would keep it as it is. The sentence at l. 331 provides the snow pit measurement dates that we refer to in the next sentences, so we would also keep it as it is.

**14.** l. 362: in "in April to June" instead of "April to May"

The sentence at l. 362 (Results) will be removed (see comment 13).

**15.** l. 370-371: Could you give the runoff in mm/d or m3/s instead of total m3? Normally, in hydrology we use either mm/d or m3/s as units for runoff, as these are easier to grasp.

This is a good remark. The runoff in April and May 2021 was $0.04 \text{ m}^3 \text{ s}^{-1}$ (or $3.1 \text{ mm d}^{-1}$) versus $1.10 \text{ m}^3 \text{ s}^{-1}$ (or $8.5 \text{ mm d}^{-1}$). We suggest expressing the runoff in $\text{mm d}^{-1}$ in the manuscript and changing the manuscript as follow:

**l. 370 to l. 372 (Results):** "*In April and May 2021, **the average runoff was 3.1 mm d⁻¹ compared to 8.5 mm d⁻¹** in the following year. The spring runoff from the low-snow and warm winter was the lowest observed at the outlet of BEREV-7A for April and May since discharge monitoring began in 1968. In 2022, it was the sixth highest.*"

**16.** l. 476: "precipitation" change to "liquid precipitation"

This will be changed in the caption of Figure 11 as follow:

**l. 375-376 (Results):** "*Figure 11: Air temperature (a–b), daily difference of SWE, **liquid** precipitation (c–d) and streamflow discharge (e–f) for winter 20-21 (left) and 21-22 (right). Air temperature and **liquid** precipitation are measured at the NEIGE site…*"

**17.** l. 389: "Their relative size shows the importance of the process between the gaps and the subcanopy locations." What do you mean by importance? Do you mean the magnitude? Please make this clear.

Indeed, we meant the magnitude. Here are the changes that we propose to make it clearer in the caption:

**l. 389 (Discussion):** "*The size **of the arrows** indicates the **magnitude** of the process **at one location relative to the others***."

**18.** l. 390: "Large black arrows are applied all three locations." This is unclear. I think you mean that this analysis is not made for the three sites but rather for the larger catchment (as data from NEIGE station and discharge gauge at outlet is used).

Indeed, the large black arrows indicate that the analysis is made at the catchment scale. Based on comment 30, we also suggest changing the color of these arrows from black to gray. We propose the following changes:

**l.389-390 (Discussion):** "*Large **gray** arrows **indicate an analysis made over the entire catchment.***"

**19.** l. 423: "with canopy closure". It is unclear to me what you mean with this, please rephrase to make it clear.

We meant that the melt rate was lower under the canopy than in the gaps, which coincides with a lower contribution of shortwave radiation to snowmelt. We suggest removing the mention to "canopy closure" and rephrasing this part of the manuscript as follows:

**l. 422 to l. 425 (Discussion):** "*The higher rate of melting in the gaps coincides with greater incoming solar radiation than under the canopy. In fact, solar radiation is known to be the main driver of melting in these environments (Malle et al., 2019; Lawler and Link, 2011; Ellis et al., 2011).*"

**20.** l. 450: For the other discussion sections you used statements as titles which makes it easy for the reader to grasp the main point. Can you also do this here?

This is a very good point. Here is the suggested title for this discussion section:

**l. 450 (Discussion):** "4.4 *Larger temperature gradient and snow permeability*"

**21.** l. 523-525: Can you elaborate on this statement. Was this expected? I would expect that in most years the precipitation and temperature conditions drive how much snow can accumulate and when it melts and this drives the spring freshet: Or are there examples where soil refreezing drives the spring freshet? Moreover, to me, it is not clear what you consider under "weather conditions" and "snow characteristic" in this context. I would think the amount of snow (SWE) belongs to snow characteristics, however, it depends on the weather conditions and influences the spring freshet.

We agree that this statement needs improvement. A frozen soil limits infiltration and favors runoff, and high snow permeability facilitates downward flow through the snowpack. Both processes would theoretically result in faster and larger spring runoff. However, our results show that the spring freshet is significantly reduced in W20-21, suggesting that low snow accumulation, early melt and low rainfall in

spring determine spring freshet. Thus, the soil thermal regime and snow properties would be of secondary importance and would affect the timing of the discharge response in conditions of low ROS and meltwater contribution. Additional observations or modeling would be needed to validate this as well as sub-daily streamflow measurements (see comment 2). We suggest the following change to the conclusion:

**l. 523 to l. 525 (Conclusion):** "***Although enhanced soil freezing and larger snow permeability point toward faster runoff and larger peak flow, our results suggest that these are of second importance as low snow accumulation in winter, early snowmelt and low spring rainfall in spring led to a significantly lower spring freshet in the warm year.***"

**Figures**

**22.** Figure 1c): It seems like a fish-eye perspective, but could you put a scale bar here, so the size is clear?

Thank you. We will add the scale bar to Figure 1c) in the revised version of the manuscript. We also suggest modifying the figure caption to identify the green area as the boreal zone in Figure 1a (see comment 23).

**23.** Figure 1a) DEM color is not so color-blind friendly

Very good point! we suggest the following revised Figure 1:

[Figure]

**l. 89 to 92 (Methods):** "*Figure 1: Map of the province of Quebec, Canada, with the location of the Montmorency Forest (MF) indicated by a red star **and the boreal zone in green** (a). Elevation map of the study catchment (BEREV-7A) with the location of the experimental site and the outlet of the catchment (b). Aerial view of the experimental site with the medium gap (yellow), the small gap (purple), and the canopy (green) stations (c) and black and white hemispherical photos of each station location (d–e–f). Picture of the monitoring station under the canopy (g).*"

**24.** Figure 2a): cumulative precipitation plots maybe better to show what you want?

We have tried different versions of Figure 2a) where we plotted the cumulative precipitation, for instance in Figure R1.2. However, showing the monthly sum of precipitation, as it was presented in the original manuscript, allows to compare both liquid and solid precipitation of winters 20-21 and 21-22 along with

the mean over the 1982-2022 period. Adding the cumulative precipitation to Figure 2a) overloads it. We prefer to keep Figure 2a) as it is.

[Figure]

Figure R1.2: Cumulative precipitation for winter 20-21(a) and winter 21-22 (b) during the snow cover period. Liquid and solid precipitation are represented by dashed and solid areas, respectively.

**25.** Figure 5: It is quite difficult to compare the two years to each other and see which values are larger, especially for c) and d). Plotting both years in a single plot would make the comparison easier for the reader.

Thank you for this very good suggestion. We have combined both years on the same frames, and it does make it much easier to read. Here is the new Figure 5 that we propose to include in the revised manuscript:

[Figure]

l. 280 to l. 282 (Results): "*Figure 5: Cumulative net total ($R_{net,bc}$), shortwave ($SWR_{net,bc}$) and longwave ($LWR_{net,bc}$) radiation below the canopy during snowmelt of winter 2020–21 (**red**) and 2021–22 (**blue**) in*

*the medium gap (**a**), in the small gap (**b**) and **at the canopy station (c)**. Graphs start at the beginning of the snowmelt period on **10 April 2021 in the first year and on 3 May 2022 in the second year**. **Note that SWR$_{net,bc}$ and LWR$_{net,bc}$ overlap during the snowmelt of 2021**.*"

**26.** Figure 6: very nice plot, very easy to grasp!

Thank you!

**27.** Figure 7: Also here it is quite difficult to compare the two years and see the differences between the two years. You can plot both years in the same plot, by using different line styles (solid, dashed).

Each panel in Figure 7 shows the ground heat flux (GHF) and the soil temperature at four depths. The difference in GHF and soil temperature between the two years is also quite small. Therefore, combining W20-21 and W21-22 on the three frames for the two gaps and the canopy makes each frame overloaded and harder to read. We would therefore prefer to leave Figure 7 as it is.

**28.** Figure 11cd: y axis label should be mm/d I guess.

The correction will be made in the revised manuscript. Also, given that we express the spring runoff in mm d$^{-1}$ in the text, it makes sense to present Figure 11e-f accordingly. Therefore, units on the y-axis of Figure 11e-f will also be changed in mm d$^{-1}$ in the revised version of the manuscript.

**29.** Figure 11: You compare the liquid precipitation over the whole catchment to the SWE averaged over the stations. Can you elaborate on whether the SWE averaged over the stations is representative of the whole catchment? Is the experimental site located at a representative elevation for the catchment?

In Figure 11, we assume that precipitation measured at the NEIGE site is representative of the precipitation at the study site. We also suppose that the amount and the phase of precipitation, as well as the SWE estimated at the study site are relatively uniform over the catchment. Given the small size of BEREV-7A (1.1 km$^2$) and since the study site is located at an elevation of 850 m ASL, which is in the mid-range elevation of the catchment (770 m to 980 m. ASL), we think that these are fair assumptions. We suggest the following addition to the method section in the manuscript:

**l. 102 (Methods):** "***Given the small size of the catchment and the location of the stations close to the average elevation of the catchment, we assume that the snow measured at the experimental site is representative of the entire catchment.***"

**30.** Figure 12: Very nice to have an overview figure of all results. A legend of what the colors mean is missing. The black arrows at temp, snowfall, precip. and discharge can be easily confused with the other arrows that just show the relationships. Using another color (maybe grey) for the large black arrows would help. Also,

Thank you, we have made the suggested changes. Here is the new version of the figure we suggest using, with the legend incorporating comments 17 and 18:

[Figure]

**l.384 to l. 391 (Discussion):** "*Figure 12: Summary of the results. Upward arrows correspond to an increase and downward arrows to a decrease in the low-snow and warm winter with respect to the reference winter. The clock with counterclockwise arrows means that the process is happening earlier. The yellow, purple, and green arrows indicate the effects in the medium gap, the small gap and under the canopy, respectively. The size **of the arrows** indicates the **magnitude** of the process **at one location relative to the others**. Large **gray** arrows **indicate an analysis made for the entire catchment.** Small black arrows show the causal link between the observations processes. Gray boxes refer to processes treated in this study (1. snowmelt dynamics; 2. soil thermal regime; 3. snow metamorphism).*"

---

## Author Comment (AC2)

**HESS-2023-191 - How does a warm and low-snow winter impact the snow cover dynamics in a humid and discontinuous boreal forest? An observational study in eastern Canada**

**Responses to the Anonymous Referee #2**
* * *
We would like to thank the anonymous referee #2 for providing relevant comments and suggestions. Based on these, we propose some changes to the original manuscript and its supplementary material. We hope that these changes, if they are accepted, will better place our study in the context of past and future climate and emphasize the originality of our work. Our answers below are in blue, whereas excerpts from the manuscript are in *italic* with modifications in **bold.**
* * *
This manuscript provides a data-rich description of two winters at Montmorency Forest (Québec, Canada) with contrasting meteorological conditions. The manuscript is interesting, easy to read and data are presented in thorough and clear manner. However, I have a number of comments that the authors may wish to consider to help guide the clarity and purpose of the messages in this paper, for the wider community.

**Major comments**

1. The authors do a good job to realize their first objective – to quantify and compare the effect of snow under forest canopy and in canopy gaps on soil properties around the phase boundary and snow properties. This creates a thorough descriptive narrative, but which very largely reinforces what we already know, and struggles to justifiably generalize beyond the study site.

    a) Snow is well known to have a very important influence on insulating the relatively cold winter air temperatures from warmer soil. Very broadly, shallow snow means cooler soils and vice versa. Slater et al. 2017 (doi:10.5194/tc-11-989-2017) demonstrated that at effective mean snow depths of 50 cm the influence of the atmosphere on soil temperatures decouples. Hence shallow sub-canopy snow at Montmorency (< 50 cm) has a bigger influence on the variability of soil temperatures relative to deeper snow in gaps.

Thank you for the comment and the reference from Slater et al., (2017). Indeed, it is well known that snow has an insulating effect on the ground thermal regime, with a thinner snowpack favoring a cooler ground. A previous study by Hardy et al. (2001) [doi:10.1023/A:1013036803050] showed that low-snow winters lead to enhanced ground freezing in a hardwood forest. Other studies (Mellander et al., 2005 [doi:10.1016/j.agrformet.2005.08.008]; Stadler et al., 1996 [doi:10.1002/(SICI)1099-1085(199610)10:10%3C1293::AID-HYP461%3E3.0.CO;2-I]) have shown that the soil is cooler and frost is deeper under the canopy than in open stands or forest gaps. Thus, both low winter snowfall and canopy interception reduce snow accumulation on the ground, thereby reducing the insulating effect of the snowpack. However, both factors (low-snow conditions and canopy cover) have not been thoroughly investigated together in a single study. We address this in our observational study and we show that the difference in ground thermal regime is greater between gaps and subcanopy than between low-snow and normal winter conditions. We suggest a restructuring of the introductory paragraph dealing with these issues:

**l. 42 to l. 55 (Introduction): "*It has been shown that the ground thermal regime is strongly influenced by the amount of snow accumulation (Zhang et al., 2005; Slater et al., 2017). In forests, the spatial pattern of soil temperatures is difficult to determine because snow depth is highly variable (Mellander***

*et al., 2005). Observations from a subalpine forest plot in Switzerland show that frost penetrates the ground deeper under tree crowns due to less snow accumulation than in forest gaps which reduces the infiltration and increases surface runoff (Stadler et al., 1996). Infiltration is also limited during low-snow winters due to a thinner snowpack that favors soil freezing (Hardy and al., 2001; Shanley and Chalmers, 1999). It is clear that both canopy structure and snow conditions influence the ground thermal regime, soil freezing and infiltration. However, it is not well understood which of these two factors predominates over the other because they have not been investigated simultaneously in a single study."*

We also propose the following addition to the Discussion:

**l. 438 (Discussion):** "*Our results are also consistent with Slater et al. (2017) as the effect of air temperature on ground temperature is more pronounced when the snowpack is shallow, which is specifically the case under canopy throughout W20-21.*"

    b) The snow properties (effective conductivity) go a little way to mediating this influence, but are secondary in importance to the magnitude of the snow depth.

We disagree that the effective thermal conductivity of snow ($k_s$) is less important than snow depth for insulating the ground. In fact, snow height and $k_s$ are rather of the same importance with regard to the thermal insulance of the snowpack (see Eq. 3 from Barrere et al. (2017)[doi: 10.5194/gmd-10-3461-2017]). The cooler ground under the canopy than in the gaps suggests that differences of snow height between the sites is greater than the difference of $k_s$. Unfortunately, we did not have simultaneous measurements of $k_s$ at the same height in different the locations to quantify the contribution of $k_s$ for insulating the ground.

    c) In addition, earlier snowmelt meaning slower melt rates due to lower incoming shortwave just reinforces the point made by Musselman et al 2017 (as cited) and the lower SWE leading to lower stream discharge is intuitive.

Indeed, but our study allows us to quantify these processes with in-situ observations, in addition to shedding light on how these processes operate in two sub-environments of the humid boreal forest, i.e., gaps and subcanopy areas. This level of analysis at high temporal resolution, covering both snow and soil processes, has rarely been undertaken.

2. While comparisons between snow and soil properties in sub-canopy and forest gaps are consistently made, the explanation of these differences is often missing and makes the discussion highly speculative. This appears in the discussion section where the language used often relies on 'would imply', 'seemed to', 'may have', 'could be' or 'suggests that' to dilute the strength of conclusions that can be drawn.

    a) An atmosphere-forest-snow-soil model would allow the quantified explanation of processes that govern the observed snow/soil properties outcomes. While I respect the authors right to control the narrative, which is currently clearly expressed that this is an observational case study, the lack of a modelling approach hugely limits the capacity to quantifiably explain key processes and help to generalize beyond this catchment and beyond the two winters presented. In particular, the capacity to explore the forest canopy impact on interacting energy and mass balance processes (second objective on ln 71) would be unlocked. You have presented a fantastic snow and soil dataset, you have good forest canopy structure data using HPEval, so

by including some process modelling (e.g. CRHM for hydrology or Crocus with a canopy model for snow properties) you would be able to more adeptly justify your explanations.

We agree that the addition of modeling would allow to generalize some of our results beyond what we found at the study site. We did indeed run simulations with a snow model (SNOWPACK version 3.6.0, see Bartelt and Lehning (2002) [doi:10.1016/S0165-232X(02)00074-5]) at this experimental site and at another site where we have a similar experimental setup. We decided not to include them for two main reasons:

- SNOWPACK (like many other snow models) uses a big-leaf approach to characterize the canopy, which means that the vegetation is represented as a homogeneous layer regardless of the structure of the canopy. This representation is not suitable for simulating snow in the gaps, which obeys different processes than under the canopy. This limitation would have prevented us from contrasting gap environments with areas under the canopy. Model improvements were found to be necessary for meeting the high level of details of our observations.
- Second, adding modeling development and model runs to our study would have greatly increased its already rich content, as we have 3 sites that are compared over two winters.

We have therefore decided to save our simulation results for a forthcoming manuscript that will focus exclusively on subcanopy environments.

It was also pointed out by the anonymous referee #1 that the second objective was not properly addressed in our study. Therefore, we have decided to drop this objective and address only the first one. We suggest the following modifications to the manuscript:

**l. 69 to l. 76 (Introduction):** "***The main research gap that motivates our work is that winter weather conditions and canopy structure have not been studied together to see how they influence snowmelt dynamics, the ground thermal regime, and the physical properties of the snowpack. Thus, the objective of this study*** *is to quantify the effect of a low-snow and warm winter* ***on the aforementioned processes*** *in a humid and discontinuous boreal forest. To assess this, we compared snow melt, snow physical properties, soil freezing, and spring runoff at a small catchment* ***in the south part of the*** *boreal forest of eastern Canada, for two consecutive winters. One winter was exceptionally warm and dry, while the other was colder, with precipitation amounts similar to the standard climatology of the study region. These contrasted conditions represent an ideal comparison to investigate some expected effects of climate change. Extensive snow monitoring and pit measurements were conducted to* ***achieve the research objective***."

**l. 492 (Discussion):** *As the conclusions of this study are based solely on observations, it would be interesting to pursue analyses with models simulating water and energy exchanges at the atmosphere-forest-snow-soil interface. Multilayer snow models such as SNOWPACK (Bartelt and Lehning, 2002) or Crocus (Vionnet et al., 2012) would be good tools in that regard, as they are able to estimate water transport in a one-dimensional snow column and the infiltration into the soil.*"

We also suggest removing sentence from l.17 to l.19 in the abstract.

For example:

b) ln 421-422 states earlier melt onset suggests net radiation is lower, but this is not shown and the impact of sensible heat fluxes are not considered? Even in the sub-canopy where turbulence is lower, when the air temperatures go above zero then sensible heat fluxes can have a significant effect.

Thank you for this interesting remark. Indeed, turbulence is low under a forest canopy and turbulent fluxes are difficult to measure under these conditions (Reba et al., 2009 doi:10.1029/2008WR007045). Unfortunately, mainly for logistical reasons, we do not have such measurements. In the other hand, turbulent fluxes are also challenging to model under canopy and in forests gaps because the surface heterogeneity and discontinuity from forest edges make the Monin-Obukhov similarity theory invalid (Conway et al., 2018) [doi: 10.1175/JHM-D-18-0050.1]. However, we acknowledge that sensible heat flux can be significant under the canopy even without turbulence and therefore suggest the following changes to the manuscript:

**l.421-422 (Discussion):** "*An earlier melt onset implied that net radiation was lower in W20–21* **(Fig. 5), which is consistent with a lower melt rate in that year (Fig. 6). Although it was not measured in this study, sensible heat flux may have contributed to snowmelt in gaps and under the canopy. This should be addressed in future modeling studies despite challenges in simulating turbulent fluxes in discontinuous forests (Conway et al., 2018).**"

> c) On ln 428-429 you state that the structure of the canopy must be considered in models. I fully agree, and here but you have the capacity to show this in a model and address your own statement.

This is a good point. A model like FSM2 (see Mazzotti et al. (2020) [doi: 10.1029/2020WR027572]) has a detailed description of the canopy structure. This model can eventually be used with our measurements but the canopy structure at our study site should first be mapped to achieve accurate simulations with FSM2. We suggest the following change to the manuscript to make it less speculative:

**l.428 to l. 430 (Discussion):** "*Overall,* **our results show that snowmelt dynamics are highly variable at the local scale in forests due to the discontinuous canopy structure. It underscores the importance of using high-resolution canopy structure mapping in snow models to accurately** *predict* **snowmelt in forests.**"

> d) A modelling approach would go some way to explain why heat loss was sufficient to favor soil freezing under a canopy but not in gaps (ln 434-435) – I would expect net LW to be important here, but comparison of modeled fluxes would allow a more robust analysis.

The ground freezes under the canopy because the heat lost to the snowpack is not compensated by the heat flux from the lower soil layers. Measurements of soil thermal conductivity ($k_{soil}$) made in the summer of 2021 at MF under the canopy with a Hukseflux TP02 heated needle probes give a $k_{soil}$ of roughly 0.8 W m$^{-1}$ K$^{-1}$ (see comment 5). The temperature difference between depths of 5 and 20 cm is 1°C ± 0.2 °C from December to February both under canopy and inside forest gaps. We can therefore estimate an average heat flux entering the topmost 5 cm of soil from beneath between 4.3 and 6.4 W m$^{-2}$. This is larger than the ground heat flux at the soil-snow interface in the gaps, but lower than under canopy. Therefore, the topmost soil layers freeze in under the canopy but not in the gaps. We propose the following addition to the manuscript:

**L. 434 (Discussion):** "*Our observations show that the heat loss was sufficient to favor soil freezing under the canopy, but not in the forest gaps.* **The topsoil thermal conductivity was measured at 0.8 W m$^{-1}$ K$^{-1}$ in the summer 2021 at the canopy station (Fig. S2). Given that the temperature difference between depths of 5 and 20 cm in the soil varies between 0.8 and 1.2 °C, we can readily estimate from Fourier's law applied to the top 5 cm of soil (Equation 5) that the average ground heat flux is 5.3 W m$^{-2}$. This is lower than the estimated snow heat flux under the canopy (Fig. 7e-f) which explains why the topmost subcanopy soil layers froze in both years.**"

e) If you couldn't manually assign an albedo class, which would be hard to do for purposes of spatial and temporal generalization, how would a modeled estimate of albedo affect the relative impact of energy and mass balance processes?

Indeed, such comparison would be interesting but unfortunately this goes beyond the scope of our study.

These are just a non-exhaustive number of examples where I feel inclusion of a simple modelling approach would allow the speculative areas of the discussion to be either removed or better justified.

In order to be coherent with our narrative, we would need a snow model that includes both a multi-layer description of the snowpack and a detailed representation of the canopy structure, which unfortunately, does not exist at the moment. Please see our response to comment 2a.

3. Much is made of winter 20-21 being analogous to a warmer climate, but while we should expect warmer winters in a warming climate, it is much less understood as to the impact on winter precipitation. This would have an impact on the mass of snow (see earlier comment) and the potential for rain on snow. Consequently, this could benefit from a much more robust underpinning using future model projections of climate (e.g. NA-CORDEX) to show not just where 20-21 fits within past measurements (nicely shown in Table 4), but where it lies in the future. Some big statements are currently being made (e.g. ln 464-465) about how snow thickness could override impacts of increased air temperatures. This could benefit from a more solid foundation in future climate projections.

Overall, the winter precipitation in eastern Canada is expected to increase in the future (Guay et al., 2015) [doi:10.1080/07011784.2015.1043583]. At the Montmorency Forest, we expect a slight increase of 4% in winter precipitation (ClimateData.ca, 2023) [https://climatedata.ca/, location Lac-Bureau). However, we also expect extreme winters, such as warm and low-snow winters, to be more frequent (Ouranos and MELCCFP, 2022) [https://cehq.gouv.qc.ca/atlas-hydroclimatique/]. This makes the warm and low-snow winter of 2020-21 at Montmorency Forest an interesting study case that illustrates a likely scenario. In order to address the two winters in this study relative to future climate, we suggest the following addition to the manuscript:

l. 238 (Results): "*In comparison, the projected DJF temperature and the total JFMA precipitation are expected to be –10.4 °C and 425 mm, respectively, by 2070 at the Montmorency Forest (ClimateData.ca, 2023).*"

Other changes were suggested related to the exceptionally warm and dry winter of 2020-21 in the context of the past and future climate. Please, see comment 1 of the document of responses to the anonymous referee #1 for the changes that we proposed.

4. The second objective features much less in the manuscript and is much more speculative (in its current form without hydrological modelling). It may be that this could be sacrificed in a revised manuscript in order to improve the focus on objective one?

We suggest dropping the second objective of the study (see comment 2a).

**Minor comments**

I appreciate the main thrust of my major comments (to include a modelling component) is non-trivial. Hence for now I will restrict the minor comments to a few obvious changes (also because the manuscript is well written and has very few obvious minor issues):

5. Ln 310-311: Are the soil profile characterizations (inc. porosity) shown anywhere? These may be important when considering soil hydraulics and thermal transfer.

Thank you for your comment. We do have soil profile characterization data. We suggest presenting them as supplementary material (Fig. S2) and introducing them as follow at the end of section 2.2.1.

**l. 138 (Methods):** "*We performed detailed soil profile measurements at the canopy and small gap sites on 13 and 20 July 2021, respectively. At the canopy site only, we measured soil thermal conductivity at different depths using a Hukseflux TP02 heating needle probe. Soil temperature was measured every 5 cm from the surface to 30 cm below and every 10 cm down to 80 cm below the surface with a Greinsinger Pt-1000 temperature probe (resolution: 0.1°C). Soil cores ($\approx$165-cm$^3$) were taken from each layer, which were then weighted before and after oven drying for 48 hours at 65 °C and 100 °C for organic and mineral soils, respectively. This allowed to estimate the volumetric water content and the bulk density of the soil, assuming a density of water of 1000 kg m$^3$. Figure S2 presents the vertical profiles of soil characterization at both locations.*"

[Figure]

"*Figure S2: Soil profile characteristics measured at Montmorency Forest on 13 and 20 July 2021 at the canopy and the small gap stations, respectively. The plots include vertical profiles of thermal conductivity (at the canopy station only), volumetric water content, bulk soil density and temperature. The organic matter layer is 7-cm thick under the canopy and 9-cm thick in the small gap.*"

6. Ln 52; delete '17-19 May'

This reference was removed in the revised version of the manuscript that we suggest (see comment 1a).

7. Ln 168: 'measurement' rather than 'measurements'.

Thank you, this typo will be corrected.

8. Ln 272: April rather than Avril.

This will also be corrected.

9. What evidence is there for the forest being humid? Particularly in the winter? No humidity measurements are presented or referred to.

This is a good remark. We described the boreal forest of eastern Canada as being humid based on the water availability index map (see Trabucco et al., (2019) [doi:10.6084/m9.figshare.7707605.v3], Figure 1a from D'Orangeville et al., (2015) [doi:10.1126/science.aaf4951]) and the work from Isabelle et al. (2020) [doi:10.1016/j.agrformet.2019.107813]. We suggest removing the sentence at lines 36 to 38 and changing to the manuscript as follow (see also comment 1a and 7 from the referee #1):

**l. 38 to l. 41 (Introduction):** "***Projections*** *for the boreal forest of eastern Canada**, characterized by humid and cold conditions in winter (D'orangeville et al., 2015; Isabelle et al., 2020),** point towards an increase in winter streamflow and an earlier spring freshet **with more snow accumulation in the north and less in the south (Guay et al., 2015). The interannual variability of precipitation and temperature is also projected to increase, making warm and dry winters more likely to happen in this region (MELCC, Ouranos 2022).**"*

---

## Author Response (AR2)

**HESS-2023-191 - How does a warm and low-snow winter impact the snow cover dynamics in a humid and discontinuous boreal forest? An observational study in eastern Canada**

**Responses to the Anonymous Referee #1 (round 2)**
* * *
We wish to thank anonymous reviewer #1 for commenting on the revised version of the manuscript and for providing additional insightful comments and suggestions. Based on these, we propose major changes to the Results and Discussion sections. We hope that these changes, if accepted, will place our results more clearly in context and better address the research objective. Our responses below are in blue, while excerpts from the manuscript are in blue *italics* with modifications in ***bold***.
* * *
Thank you for submitting the revised version of your manuscript. All my minor comments were well addressed in this revised version of the manuscript. Also, the major comments were mostly addressed but additional clarifications regarding my comments referring to the exceptionally dry year and the research gap are still needed.

**Major comments**

**1. Discussion on how climate change impacts would deviate from your results and conclusions**

The authors start the paper by highlighting that climate change will affect the boreal forest and that climate change impacts are difficult to assess in boreal forests because of its complexity (l.14-19). They use this as a motivation for the observational study, which will add knowledge about potential climate change impacts. At the end of the introduction you also mention the "contrasted conditions represent an ideal comparison to investigate some expected effects of climate change.".

They come back to climate change in the conclusion "Although this work is limited to a two–year comparison within a small catchment, it highlights the many potential effects, all together, of a changing climate on snow hydrology in a discontinuous boreal forest through a unique set of highly detailed process–level observations.".

You argue that due to higher variability such dry-warm years as winter 20-21 are becoming more likely. As you mentioned, however, winter precipitation is expected to increase in future. Thus, I would argue that dry-warm years will still be the exception in future. More likely are wetter-warmer years, with more rain and less snow. So, as you also acknowledged in your reply to the first revisions, winter 20-21 is not representative of the projected future. The climate change projections you added in lines 242-244 underline this. It shows that average winter precipitation is projected to be twice as high in future, compared to precipitation in winter 20-21, but the average temperature is projected to be similar to winter 20-21. Thus, such a low snow year will probably also be exceptional in future and more precipitation than in winter 20-21 will be normal.

Therefore, I think you need to mention better in your discussion that the conditions during your study resemble a condition that is more likely in future but that will not be the norm. Therefore, I think you need to add to the discussion some sentences about how your results differ compared to projected average future conditions, especially since you use climate change as a predominant motivation for your study. What would be different if winter precipitation would be higher (wet-warm year)? Would there be probably more snow than in your results? More ROFs? How would this affect the processes you discuss?

As mentioned by the reviewer, since the projected increase in precipitation will be combined with an increase in temperature, it is expected that liquid precipitation will increase in winter at the expense of solid precipitation. Therefore, the number of days with snow and the maximum snow accumulation in winter will likely decrease (see Fig.8 from Guay et al. (2015) [doi:10.1080/07011784.2015.1043583]). Although the low precipitation that we observed in W20-21 is not consistent with the climate change projections, the observed low snowfall and low snow accumulation on the ground in W20-21 is. This should be the main justification for considering W20-21 as representative of future winters, complemented by the interannual meteorological variability.

With respect to wet and warm winters, we expect an increase in the likelihood of ROS events and a decrease in solid precipitation. Therefore, the snowpack thickness would be less than in a normal winter. Since the snowpack thickness exerts a control on the ground thermal regime and on snow metamorphism, it is reasonable to expect that the processes described in this manuscript would not differ much in a warm and wet winter. However, this remains speculative and a third winter of observations (warm and wet) would be needed to confirm this hypothesis.

Overall, we agree that W20-21 should be placed more clearly in the context of climate projections for eastern Canada, based on temperature and snow cover duration and thickness. Therefore, we propose the following changes to the manuscript:

**l. 14 to 15 (Abstract):** "*In the boreal forest of eastern Canada, winter temperatures are projected to increase substantially by 2100. **This** region is also **expected to receive less solid** precipitation, resulting in a reduction in snow cover thickness and duration.*"

**l. 439 to 446 (Discussion):** "*In eastern Canada, as in other high-latitude and high-elevation regions, the snow cover extent is expected to decrease due to warmer winter temperatures (Guay et al., 2015; Pepin et al., 2015; Kunkel et al., 2016). **This region is also expected to receive more winter precipitation in the future (Guay et al., 2015; Ouranos and MELCCFP, 2022). Therefore, the exceptionally warm and dry conditions observed at MF in W2020-21 are not entirely consistent with the median climate projections for eastern Canada. However, these conditions did result in a snowpack that melted out 23 days earlier and in a maximum snow height that was 36% lower than the 1982-2022 reference period (Table 4). Based on these low snow accumulation conditions, the winter of 2020-21 is representative of what can be expected in eastern Canada with climate change even though the expected more abundant liquid precipitation may lead to more significant snowpack modifications than observed in W20-21.**"*

**2. Research Objective**

The revised introduction is well written. As main research gap that motivates your study, you mention that weather conditions and canopy structure have not been studied together to see their influence (l. 62-64). Before in the introduction you mention that it is not well understood "which of the two factors predominates because they have not been investigated simultaneously in a single study" (l.57-58).

Although the objective of the study is much clearer in this version of the manuscript, I do not have the feeling you addressed clearly which factor (canopy structure or weather) predominates in the investigated processes. To fully address the research gap a paragraph could be added to the discussion which discusses the two factors (weather and canopy structure) and the relationship between them that you can draw/hypothesize based on your results, e.g. what affects the snow dynamics, the ground thermal regime, and the physical properties of the snowpack more, the complex structure of the canopy or the climate conditions?

Thank you for this very good suggestion. Based on this, we decided to compare canopy and gap locations using several parameters: maximum snow height, snow surface temperature, melt-out date, ground heat flux, ground temperature, vertical temperature gradient and density, SSA and snow permeability profiles. This comparison shows that the meteorological forcing exerts a stronger control on the snow accumulation and melt dynamics, that forest structure controls the freezing of the top-most soil layers and that both meteorological forcing and forest structure produce significant differences in the snowpack physical properties. We propose the following additions to the manuscript in the Abstract and in the Results and Discussion sections:

[revised manuscript text omitted]

**Minor Comments**

3. l. 242-244: It's good that you mention what temperatures and precipitation are expected. However, it is also important to mention which emission scenario is used for generating these projections and whether these are ensemble projections of multiple GCMs.

Thank you for that observation. We used the SSP5-8.5 emission scenario based on the CMIP6 climate simulations to obtain the temperature and climate projections to 2070 at our study site. We suggest the following change to the manuscript:

**l. 263 to 266 (Results):** "*In comparison, the projected DJF temperature and the total JFMA precipitation are expected to be –10.4 °C and 425 mm, respectively, by 2070 at the Montmorency Forest, **based on the SSP5-8.5 emission scenario from CMIP6 climate simulations** (ClimateData.ca, 2023).*"

**HESS-2023-191 - How does a warm and low-snow winter impact the snow cover dynamics in a humid and discontinuous boreal forest? An observational study in eastern Canada**

**Responses to the Anonymous Referee #2 (round 2)**
* * *
We would like to thank the anonymous reviewer #2 for complementing on his/her previous comments. Based on these new comments, we plan to make significant changes to the manuscript. First, we propose to deepen the comparison with the study by Slater et al. (2017) in the Discussion. Second, we also propose to include SNOWPACK modeling results in the manuscript, which we hope will provide a clearer explanation of the observed processes. Our responses below are in blue, while excerpts from the manuscript are in *italic* with changes in ***bold.***
* * *
I would like to thank the author for making changes to their manuscript in light of the two reviews. In particular, removing the second objective from the original manuscript has helped improve the focus of the study. There are two outstanding comments that I feel would improve the paper.

**Major comments**

1.  The addition of text using Slater et al. (2017) is welcomed. However, your response that effective conductivity is not less important than snow depth, is not shown by Figure 3 in Slater et. al. (2017). The relationship is dominated by snow depth, while the noise around it is likely due to snowpack properties. If snow properties were more important than snow depth, then there would be no relationship of this sort. Arctic tundra snowpacks, like Bylot Island (Barrere et al. 2017, 10.5194/gmd-10-3461-2017) where proportions of wind slab and depth hoar play a very important role in bulk thermal conductivity, will contribute to the variability around the relationship in Slater et al. (2017). However, the range of latitudes used by Slater et al. (2017), see Fig 2, aggregates lots of lower latitude snowpacks of the type expected in Montmorency Forest. I would expect snowpacks of the type measured in Montmorency Forest to be more closely related to snow depth. It would be nice if the authors could consider adding the gist of this to their discussion so that the implications of their work in Montmorency can be put in context of forest snowpacks in general.

As noted by Slater et al. (2017), the significant scatter in the data points in Fig. 3 could be due to climate variability, snow metamorphism patterns, soil moisture properties and measurement uncertainties. We agree with the reviewer that the effective thermal conductivity of snow ($k_s$) could contribute to the scatter, while maintaining an exponential increase in the normalized air-soil temperature difference. However, Slater et al. (2017) do not quantify the role of $k_s$ in their model of snow thermal insulation. Also, the study does not integrate data points from the boreal region of eastern Canada, which is characterized by cold and humid conditions. However, we believe it is insightful to apply the methodology of Slater et al. (2017) to our winter site and compare our observations to their dataset:

[Figure]

Fig. R1: Figure 3 from Slater et al. (2017) showing the observed relation between normalized temperature amplitude difference ($A_{norm}$) and the effective mean snow depth along with the resulting exponential fit (dashed line) and our data points. The grey shading shows the median fit plus/minus the mean scatter of all fits. Red dots show the warm and low-snow winter while white dots show the reference year.

Our data points fall within the scatter of Fig. 3 from Slater et al. (2017) and follow the observational fit shown in the figure. Our data points cover a large range of effective mean snow depth, but a much smaller range of normalized temperature amplitude difference ($A_{norm}$). This suggests that even in an exceptionally low-snow and warm year, the air and soil temperatures in a site like the Montmorency Forest (MF) are decoupled under the canopy and within forest gaps. This further supports our observations that weather conditions have only little effect on the soil thermal regime. Overall, we suggest the following changes to the manuscript:

**l. 494 to 502 (Discussion):** "*Slater et al. (2017) assessed the influence of snow depth on the decoupling of seasonal air and soil temperature amplitudes from multiple locations in the Northern Hemisphere. Although this study does not include sites from eastern Canada, we chose to compare our results with the observations presented in Fig. 3 from their paper. We observed that air and soil seasonal temperature signal decouples in both years at all three locations. This supports our previous findings that the exceptional low-snow and warm conditions met in winter 20-21 had almost no effect on the thermal insulation between the soil and the air due to thickness and properties of the snow (Fig. 7; Table 5). The increased snow faceting in W20-21 may also have contributed to a more efficient insulation of the ground despite a thinner snow cover.*"

2. The original comment that "while comparisons between snow and soil properties in sub-canopy and forest gaps are consistently made, the explanation of these differences is often missing and makes the discussion highly speculative", still remains. Consequently, the suggestion to use an atmosphere-forest-snow-soil model would allow the quantified explanation of processes that govern the observed snow/soil properties, is still valid.

   While big-leaf approaches are not an ideal way to represent the forest canopy, many modeling approaches still use these successfully with two or more simple parameters to represent shortwave extinction and longwave emittance. This would still be of use. However, what would be better, would be to use the more sophisticated two-layer canopy approach of Gouttevin et al. (2015) 10.5194/gmd-8-2379-2015, which the authors have used in their paper currently under discussion in The Cryosphere (10.5194/egusphere-2023-3012) and which they allude to in their response. A twolayer forest canopy model coupled to SNOWPACK (as per Gouttevin et al. 2015) would very adequately allow the impacts on snowpack structure to be investigated in the manner suggested in the first review. The readers of this manuscript in HESS would benefit enormously from including model results which would allow greater explanation of differences between sub-canopy and forest gaps and provide stronger conclusions that could more easily be generalized

We acknowledge that the use of SNOWPACK with the two-layer canopy would provide complementary explanations for the hydrological processes studied. Although this model resolves the subcanopy longwave radiation well (Gouttevin et al., 2015, doi: 10.5194/gmd-8-2379-2015; Todt et al., 2018, doi:10.1029/2018JD028719), SNOWPACK is not parameterized to simulate snow-forest processes in forest gaps (i.e., direct throughfall, shortwave extinction, longwave emissions from the gap edges and wind speed attenuation). One could simulate snow in an open site, but this would not be representative of the gap snowpack. Implementation of HPEval (Jonas et al., 2020, doi:10.1016/j.agrformet.2020.107903) into the two-layer canopy scheme of SNOWPACK may help to improve the representation of shortwave transmission in gaps. However, this would rather require significant changes to the model. Such developments are beyond the scope of this short addition to the paper here and should be the subject of a separate study.

Given the limitations described above, we propose to use SNOWPACK with the two-layer canopy to compare the melt period of the warm/low-snow year with the reference year for the snowpack under the canopy. This allows the study of some processes and variables that are difficult to analyze with observations alone. A such, we propose to include simulations of snowpack runoff, liquid water content and basal ice layer formation to the observations presented in Figure 11. We would use the same initial canopy parameterization as presented in our paper discussed in The Cryosphere (10.5194/egusphere-2023-3012). We propose the following changes to the manuscript:

[revised manuscript text omitted]

DOI of the new references cited:

- Bouchard et al., 2024 (doi:10.5194/egusphere-2023-3012)
- Gouttevin et al., 2015 (doi:10.5194/gmd-8-2379-2015)
- Hadiwijaya et al., 2020 (doi:10.3390/f11020237)
- Isabelle et al., 2018 (doi:10.1016/j.agrformet.2018.07.022)
- Lehning et al., 2002 (doi:10.1016/S0165-232X(02)00073-3)
- Mazzotti et al., 2020 (doi:10.1029/2019WR026129)
- Mazzotti et al., 2023 (doi:10.5194/egusphere-2023-2781)
- Todt et al., 2018 (doi:10.1029/2018JD028719)
- Westermann et al., 2011 (doi:10.5194/tc-5-945-2011
- Wever et al., 2014 (doi:10.5194/tc-8-257-2014)